# REPORTBENCH: EVALUATING DEEP RESEARCH AGENTS VIA ACADEMIC SURVEY TASKS

## ABSTRACT

The advent of Deep Research agents has substantially reduced the time required for conducting extensive research tasks. However, these tasks inherently demand rigorous standards of factual accuracy and comprehensiveness, necessitating thorough evaluation before widespread adoption. In this paper, we propose Report-Bench, a systematic benchmark designed to evaluate the content quality of research reports generated by large language models (LLMs). Our evaluation focuses on two critical dimensions: (1) the quality and relevance of cited literature, and (2) the faithfulness and veracity of the statements within the generated reports. ReportBench leverages high-quality published survey papers available on arXiv as gold-standard references, from which we apply reverse prompt engineering to derive domain-specific prompts and establish a comprehensive evaluation corpus. Furthermore, we develop an agent-based automated framework within ReportBench that systematically analyzes generated reports by extracting citations and statements, checking the faithfulness of cited content against original sources, and validating non-cited claims using web-based resources. Empirical evaluations demonstrate that commercial Deep Research agents such as those developed by OpenAI and Google consistently generate more comprehensive and reliable reports than standalone LLMs augmented with search or browsing tools. However, there remains substantial room for improvement in terms of the breadth and depth of research coverage, as well as factual consistency. The complete code and data will be released publicly.

## 1 INTRODUCTION

The rapid development of LLM-powered Deep Research agents has revolutionized the process of knowledge synthesis by enabling autonomous execution of extensive research tasks, including academic literature surveys, industry analyses, and market assessments (Chen et al., 2025; Gottweis et al., 2025; Lu et al., 2024; Tang et al., 2025; Yamada et al., 2025; Zheng et al., 2025; Li et al., 2025). Tasks that traditionally required days or weeks of manual effort can now be completed within minutes. Notable examples include advanced systems such as OpenAI (OpenAI, 2025) and Google's Gemini Deep Research (Google, 2025), which effectively integrate various external tools and perform multiple rounds of deep reasoning. Despite their promising capabilities, widespread practical adoption critically depends on their ability to consistently deliver research reports with high factual accuracy and comprehensive content quality. Therefore, it is essential to monitor and ensure the quality of generated reports through evaluation. However, defining what constitutes a good report is challenging and lacks broad consensus, resulting in the current absence of mature evaluation methodologies for research report generation.

In addressing this challenge, we decompose the evaluation of research reports generated by LLMs into two core dimensions: writing quality and report content. Due to the subjectivity of writing-style evaluation, while the criteria for assessing content quality can be more clearly defined, this work focuses primarily on the evaluation of report content, leaving the assessment of writing quality to future work. Specifically, we assert that the content quality of research reports hinges on two critical factors: (1) the quality and relevance of cited literature, and (2) the faithfulness and veracity of generated statements, whether derived from cited references or produced by the model.

To establish a high-quality benchmark capable of rigorously assessing research reports, we propose ReportBench, a novel evaluation framework leveraging expert-generated literature reviews. Given the constraints of relying on human annotators, who typically vary in expertise, we propose using published survey papers available on arXiv as gold-standard references. Published survey papers are typically written by domain experts and have undergone a peer review process that provides additional expert-level validation, considered among the highest-quality research reports currently available.

In practice, our methodology unfolds in two phases. First, we generate domain-specific retrieval prompts directly from expert-authored survey papers on arXiv: by analyzing each paper's publication date and full text, we generate three granularity levels of prompts (sentence-level, paragraph-level, and richly detailed versions) that precisely capture the scope, methods, and temporal constraints of the original research. These prompts form the backbone of our evaluation corpus, ensuring that downstream agents search and synthesize information within the exact topical and chronological boundaries of each survey. We extract the list of cited references from the arXiv surveys as the ground truth. Given the synthesized prompts as test inputs, Deep Research agents conduct research and generate reports, which are then evaluated based on the reference overlap with the ground truth, serving as a measure of the research skills.

In the second phase of our validation pipeline, we design two different verification procedures based on whether a statement includes an explicit citation to external literature. Specifically, for cited statements, the system identifies all in-text citations within the report, maps each citation to its corresponding source document, and employs semantic matching to ensure factual support from the cited literature. For non-cited statements, the framework employs a voting mechanism across multiple web-connected models to verify the factuality of these statements. By combining these complementary validation procedures, ReportBench delivers a systematic and detailed assessment of AI-generated research reports, ensuring the relevance and quality of cited literature and the factual accuracy of all claims through citation-based and web-based validation. To validate these automatic metrics and better understand errors, we further conduct a human evaluation and qualitative error analysis on a subset of reports. The human evaluation results exhibit high agreement with our automatic evaluation pipeline, further confirming the quality of ReportBench.

Our contributions can be summarized as follows:

- We present **ReportBench**, a systematic benchmark designed to evaluate the quality of research reports generated by Deep Research agents, with a focus on the quality of references and the factual accuracy of all statements presented in the report.

- We propose an automated and scalable data synthesis method for constructing academic survey tasks, including prompts and ground truth, from expert-authored survey papers on arXiv. Additionally, we introduce an automatic agentic evaluation framework that evaluates the precision and recall of the generated report with respect to the ground-truth references and performs factual verification of individual claims made within the report.

- We release a comprehensive benchmark suite—datasets, prompts, and evaluation scripts—to support reproducible research and community-driven progress in evaluating LLM-based knowledge synthesis.

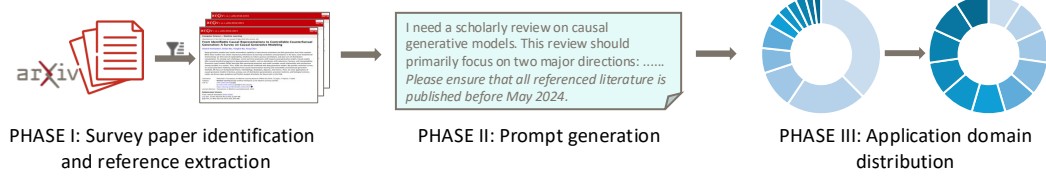

PHASE I: Survey paper identification and reference extraction   PHASE II: Prompt generation   PHASE III: Application domain distribution

Figure 1: Overall benchmark data construction workflow.

## 2 METHODOLOGY

We introduce **ReportBench**, a comprehensive evaluation framework designed to rigorously assess Deep Research agents through two interconnected components: (i) the automated construction of

high-quality benchmark datasets derived from expert-authored survey papers, and (ii) a systematic validation pipeline that evaluates the quality and factual consistency of AI-generated research reports. In the following sections, we detail the processes that underlie the synthesis of the dataset and the design of our evaluation workflow.

## 2.1 DATASET CONSTRUCTION

In this section, we detail the end-to-end pipeline to construct high-quality deep research questions along with ground-truth answers based on published survey papers. This workflow comprises three consecutive phases: (i) survey paper identification and reference extraction, (ii) prompt generation, and (iii) application domain distribution. A diagram illustrating the data construction process is presented in Figure 1.

### 2.1.1 PHASE I: SURVEY PAPER IDENTIFICATION AND REFERENCE EXTRACTION

The first step is to identify high-quality survey papers to create evaluation tasks. We start from the complete arXiv metadata snapshot (arXiv.org submitters, 2024) and retain papers submitted on or after 2020-01-01. To ensure the quality of papers, we only select those that have undergone peer review and have been formally published. We achieve this by using regular expressions, *i.e.*, querying over titles to match *"survey"* or *"review"* to filter survey papers and searching *"published"* or *"accepted"* in the comments field of a submission. To reduce systematic false positives in domains such as astronomy, we prompted GPT-4o (Hurst et al., 2024) with each paper's title and abstract to produce a binary classification of whether the paper is a literature survey.

For each survey paper, we analyze its LaTeX source file to extract cited references. Specifically, we parse LaTeX citation commands, identify and retrieve relevant bibliographic entries from associated bibliography databases, and filter these to retain only references explicitly cited in the main text. Hence, the extracted bibliography mirrors the true citation pattern of the paper. The resulting dataset constitutes a gold-standard benchmark for evaluating retrieval precision. Finally, we retained 678 papers.

### 2.1.2 PHASE II: PROMPT GENERATION

Survey papers can be regarded as a great depth of research work focused on a specific topic at a specific time, making it possible to create deep research questions in a *reverse prompt engineering* manner. In other words, given the publication date and the full text of a survey paper obtained through a PDF parsing tool, we prompt an LLM to generate a query whose ideal answer is precisely that paper. Hence, we obtain a query and its ground truth (the survey paper itself). To increase the diversity of prompts, we design three types of prompt templates:

```
Sentence-level prompt
A single sentence that succinctly defines the overarching
academic field covered by the survey.
Paragraph-level prompt
A short paragraph elaborating the research area, its main
subtopics, and the methodological perspectives covered in the
survey.
Detail-rich prompt
A detailed question that comprehensively describes the
specific research domain, key research directions, and
the methodological approaches of interest.  Additional
constraints may be included, such as preferred conferences
or journals, language of the cited literature (e.g., English,
Chinese), participating institutions or laboratories.
```

In addition, to ensure that LLMs' retrieval window matches the survey's citation horizon and prevents leakage of post-publication knowledge, we require each generated prompt to include a cut-off

date corresponding to the most recent update of the paper. For example, an expression like the following is needed.

> "Ensure only papers published before April 2025 are referenced."

Nevertheless, we still observe a phenomenon akin to prompt hacking during model evaluation, *i.e.*, the model disregards the imposed temporal constraints and directly retrieves the original source paper. As some tested systems integrate search tools internally, tool-side restrictions cannot be applied for fair comparison. To address this issue, we augment the prompt with an additional explicit instruction, stipulating that the model must refrain from citing the original paper corresponding to the given prompt. We present three prompt examples in Appendix A.3

### 2.1.3 PHASE III: APPLICATION DOMAIN DISTRIBUTION

To facilitate a more granular analysis of tested models, we classified the prompts into distinct application domains. Specifically, we utilize Gemini 2.5 Pro (Comanici et al., 2025) to classify each paper based on the title and abstract. This process yields ten distinct categories, as shown in the following box. To reduce misclassification, we introduce an unknown category, allowing the model to assign uncertain cases to this class.

| | | | |
|---|---|---|---|
| **A** | Basic Research and Scientific Exploration | **F** | Transportation and Smart Mobility |
| **B** | Information and Communications Technology | **G** | Public Safety and Social Governance |
| **C** | Artificial Intelligence and Data Intelligence | **H** | Finance and Business Services |
| **D** | Healthcare and Biomedicine | **I** | Energy and Environmental Sustainability |
| **E** | Manufacturing and Smart Manufacturing | **J** | Culture, Media, and Digital Content |
| **K** | Unknown Category | | |

The distribution of prompts across these domains is inherently biased due to the specific disciplinary focus of the arXiv corpus, as shown in Figure 2. From the pool of 678 candidate surveys, we assign each survey to an application-domain bucket and perform stratified random sampling so that domains contribute roughly equally. To create a balanced and general test set, we sample a fixed number of surveys per domain (10 in our implementation) using uniform random sampling with a fixed seed (random seed = 42), yielding 100 tasks in total without any additional hand-picking beyond the earlier filters. As we have mentioned before, we create three types of prompts for each paper. Thus, we randomly sample from these three types to obtain the final prompt with diversity. In other words, a dataset with 100 prompts is created, which we name **ReportBench**. The quality of the classification of this subset was then reviewed and validated by four research experts.

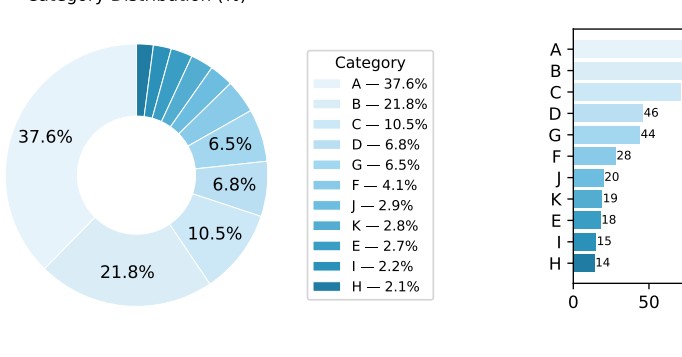

(a) Category distribution (pie).  (b) Category counts (bar).

Figure 2: Application domain distribution of the 678 filtered ReportBench prompts: (a) a pie chart showing the proportion of each application domain, (b) a bar chart illustrating the total task counts across all 11 categories.

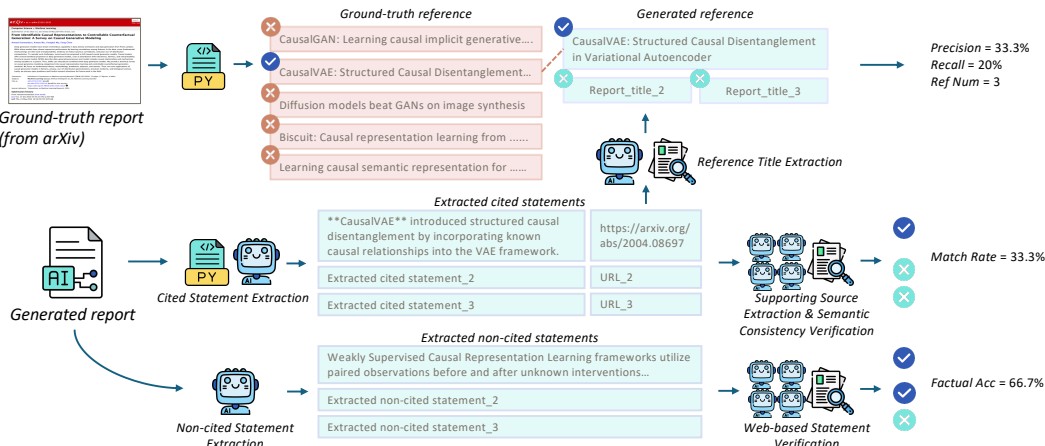

Figure 3: Evaluation Process.

## 2.2 EVALUATION PROCESS

Our evaluation process, as shown in Figure 3, uses test prompts derived from reverse prompt engineering, which require models to generate complete research reports under two constraints: a time limit and a restriction against referencing the original report. **Content quality** is first evaluated by assessing the cited references: we compare the reference list in the generated report with that of the ground truth, and the overlap ratio between the two lists serves as an indicator of the report's overall quality. Note that the time limit is enforced via the ground-truth bibliography: for each task we only include references published on or before the source survey date and exclude the survey itself, so any citation to post-cutoff papers or to the source survey lies outside the ground-truth set is counted as an error, automatically lowering precision/recall even if the model ignores the instruction. **Statement factuality** is further assessed through two complementary validation procedures. For cited statements, we verify alignment with source documents via semantic matching, while for non-cited statements, we adopt a multi-model voting mechanism to assess factual correctness. This dual strategy ensures both the faithfulness of cited content and the veracity of non-cited claims in evaluating Deep Research reports. Prompts for evaluation are presented in Appendix A.4.

**Content Quality.** We first extract all URLs from the report. Since most reports generated by the Deep Research products use URL links to cite web pages, we adopt the same citation format throughout our evaluation, including when assessing the base models. While this approach results in longer text, it offers the advantage of placing the citation immediately adjacent to the corresponding statement, which ensures consistent performance even under chunked evaluation settings. After normalizing and deduplicating them, we retrieve the content of each web page. An LLM is then used to determine whether each page corresponds to a scholarly article and, if so, to extract the article title. Finally, we compute the overlap between the extracted document titles and the ground-truth reference titles to produce a quality score.

In our current implementation, we normalize and match URL-style citations, as models are instructed to cite using URLs and the released evaluator assumes this format. This is a design choice rather than a fundamental limitation: the same extraction–normalization pipeline can be extended to BibTeX entries, arXiv IDs, and DOIs by adding format-specific extractors and mapping them to canonical URLs (e.g., https://doi.org/..., https://arxiv.org/abs/...) before comparison with the ground-truth bibliography, enabling systems to use their native citation styles without changing the evaluation protocol.

**Cited Statements.** We design a three-stage structured validation pipeline. First, an LLM automatically identifies all statements in the generated report that contain explicit citation links, establishing a mapping between each statement and its referenced source. Second, we retrieve the full content of each cited webpage via web scraping and prompt the LLM to locate the most semantically relevant passage that supports the original statement. Finally, the LLM performs consistency verification by

comparing the statement with the retrieved content, and the results are aggregated to compute an overall citation consistency score for the report. Unlike traditional "LLM-as-a-judge" approaches, which often suffer from instruction-following issues or biased scoring, our method decomposes the evaluation into fine-grained, interpretable, and verifiable steps. All intermediate outputs are retained for optional human inspection, thereby maximizing the reliability and transparency of the evaluation process.

**Non-cited Statements.** We use a simple two-step validation process. First, we extract all factual statements in the report that do not have any citations, and remove content that is general common sense or already supported by references. Then, we ask several web-connected LLMs to check each statement independently. Each model looks up information online and gives its judgment. We combine their answers using a voting mechanism to decide whether the statement is likely to be correct. This approach avoids relying on a single model and makes the validation more reliable.

## 3 EXPERIMENT

In this section, we present the performance of a diverse set of models evaluated on ReportBench. Specifically, we examine specialized Deep Research agents from OpenAI and Google Gemini. Additionally, we assess several state-of-the-art (SOTA) base models, originally lacking native Internet access, by augmenting them with an external search engine and link reader to enable the web-retrieval capabilities essential for completing our evaluation tasks. These enhanced base models are then benchmarked alongside the native Deep Research agents.

### 3.1 SETTINGS

Our evaluation pipeline uses different LLMs for distinct components. For statement extraction, supporting source extraction, and semantic consistency verification, we adopt GPT-4o. For the fact-checking of non-cited statements, we employ two web-connected models: Gemini-2.5-Pro and Gemini-2.5-Flash. Each model performs three independent judgments per statement, resulting in a total of six verdicts. The final decision is determined by majority voting, and the proportion of votes is recorded as a confidence score. In the evaluation of base models, we integrated search and link-reading tools using each model's native function call interface. Specifically, we used SerpAPI[1] for Google Search access and Firecrawl[2] for retrieving web pages in Markdown format. Due to context length limitations, we capped the maximum number of tool calls at five per instance.

To evaluate the performance of both Deep Research agents and base models, we manually collected responses from the web-based interfaces of OpenAI and Gemini, as well as batch-executed outputs from the base models, during the time window from July 14–25, 2025. Hence, the present results correspond to the July 2025 snapshot. During data collection, we ensured that OpenAI was using the standard version of Deep Research, powered by the o3 model. For Gemini, we made sure that both the "Gemini 2.5 Pro" and "Deep Research" toggles were enabled on the web interface to activate its full research capabilities.

### 3.2 EVALUATION METRICS

As described in our evaluation logic, we define three sets of metrics to assess a model's performance in conducting scientific research tasks. First, we compute the **precision** and **recall** of retrieved references against the ground-truth references. Precision reflects the proportion of cited references that are relevant, while recall measures the proportion of ground-truth references successfully retrieved. We also report the average number of references per report to capture the model's reference density. To evaluate statement-level performance, we measure the average number of cited statements and non-cited statements per report. For cited statements, we compute the **match rate**, i.e., the proportion of statements that are semantically consistent with their cited sources. For non-cited statements, we compute the **factual accuracy**, defined as the proportion of statements that are verified to be factually correct via web-connected LLMs.

---

[1] https://serpapi.com/
[2] https://www.firecrawl.dev/

| Test Model | Reference | | | Cited statements | | Non-cited statements | |
|---|---|---|---|---|---|---|---|
| | Precision | Recall | Ref Num | Match Rate | Count | Factual Acc | Count |
| OpenAI Deep Research | **0.385** | 0.033 | 9.89 | **78.87%** | 88.2 | 95.83% | 38.9 |
| Gemini Deep Research | 0.145 | **0.036** | 32.42 | 72.94% | 96.2 | 92.21% | 49.6 |
| Gemini-2.5-Flash | 0.237 | 0.012 | 5.47 | 44.88% | 12.1 | **98.52%** | 11.5 |
| Gemini-2.5-Pro | 0.269 | 0.010 | 4.27 | 59.24% | 6.58 | 96.08% | 9.35 |
| o3 | 0.299 | 0.031 | 12.26 | 31.43% | 16.16 | 82.22% | 11.51 |
| Claude-4-Sonnet | 0.337 | 0.021 | 6.74 | 73.67% | 14.93 | 92.64% | 17.07 |

Table 1: Performance metrics of OpenAI Deep Research, Gemini Deep Research, and the base models. "Ref Num" denotes the average number of references per report, and "Count" denotes the average number of cited or non-cited statements.

### 3.3 PRODUCT-LEVEL COMPARATIVE ANALYSIS

Table 1 presents the performance metrics of OpenAI Deep Research and Gemini Deep Research. In terms of retrieval performance, OpenAI achieves significantly higher precision (0.385) compared to Gemini (0.145), indicating that the references it retrieves are more likely to match the gold-standard set. Gemini shows a slightly higher recall (0.036 vs. 0.033), but this gap is negligible in practical terms. As shown in the table, Gemini generates over three times as many references per report (32.42 vs. 9.89), yet this increase does not translate into a significant improvement in recall. This suggests that Gemini tends to over-generate citations without proportionally improving the coverage of high-quality references. In some cases, excessive citation may even introduce redundancy or dilute the relevance of retrieved content. Given that the ground truth from ReportBench includes an average of 153 references per paper, with many citations supporting the same or overlapping statements, we believe recall should be considered a secondary signal rather than the primary focus of evaluation.

In terms of statement quality, both products demonstrate strong performance in generating reports. OpenAI Deep Research achieves a higher citation match rate than Gemini (78.87% vs 72.94%) while producing 88.2 cited statements on average, suggesting stronger precision in citation usage. For non-cited statements, Gemini produces more such content (49.6 vs. 38.9), while OpenAI achieves better factual accuracy (95.83% vs. 92.21%), indicating its stronger calibration in generating reliable citation-free content.

Our evaluation pipeline relies on automatic URL-to-paper mapping and LLM-based judgments of citation correctness and factuality. To assess the reliability, we also conduct a small-scale human expert study to validate our automatic evaluation pipeline and find high agreement (typically 84–96%) between expert judgments and our URL mapping, statement-level factuality checks, and citation-level precision/recall metrics, indicating that the pipeline is well-aligned with domain experts. For more details, please refer to Appendix A.2.3.

### 3.4 MODEL-LEVEL COMPARATIVE ANALYSIS

We now analyze the results across several foundation models and compare them with the corresponding Deep Research agents.

**OpenAI Deep Research vs. o3**
OpenAI Deep Research and o3 exhibit similar retrieval performance, with precision (0.385 vs. 0.299) and recall (0.033 vs. 0.031) showing only slight differences. Meanwhile, the average number of references per report is also comparable (9.89 vs. 12.26). This observation aligns well with OpenAI's official disclosure that the retrieval and synthesis backbone of Deep Research is powered by the o3 model (OpenAI, 2025).

However, we observe substantial differences in the number and quality of generated statements. OpenAI Deep Research produces significantly more cited statements on average (88.2 vs. 16.16) and more non-cited statements (38.9 vs. 11.51), while achieving a notably higher citation match rate

(78.87% vs. 31.43%) and factual accuracy (95.83% vs. 82.22%). This suggests that Deep Research is not a direct output of `o3`, but rather likely incorporates an additional writing module, possibly optimized via fine-tuning or structured pipelines. Such a pipeline may be responsible for structuring retrieved content into a more coherent, citation-aligned report.

**Gemini Deep Research vs. `Gemini-2.5-Pro`**
Similarly, Gemini Deep Research and its base model `Gemini-2.5-Pro` diverge significantly across multiple dimensions. Gemini Deep Research trades off some precision (0.145 vs. 0.269) to achieve much higher recall (0.036 vs. 0.010) and generates far more references per report (32.42 vs. 4.27). In terms of statement volume, it produces many more cited statements (96.2 vs. 6.58) and non-cited statements (49.6 vs. 9.35). Despite this increase in volume, its citation alignment remains strong (72.94% vs. 59.24%), while its non-cited statement accuracy is slightly lower than the base model (92.21% vs. 96.08%). These pronounced gaps—in precision/recall trade-off, citation count, and overall coverage—mirror the contrast observed between OpenAI Deep Research and `o3`, and suggest that the system has undergone targeted optimization for thorough research and report generation. Taken together with the visible "plan" and "step-by-step reasoning" phases presented in the Gemini Deep Research web interface, it seems plausible that the system functions more like a thoughtfully constructed multi-agent workflow or pipeline.

**Base-Model Comparison**
Among the four base models, `Claude-4-Sonnet` demonstrates the most balanced performance—achieving a precision of 0.337, a recall of 0.021, an average of 6.74 reference documents per report, a high citation semantic consistency (73.67%), and a strong non-cited statement factual accuracy (92.64%). In contrast, `Gemini-2.5-Pro` attains higher precision (0.269) at the expense of recall (0.010) and generates fewer reference documents on average (4.27 per report), limiting its coverage. `Gemini-2.5-Flash` underperforms on both precision (0.237) and recall (0.012), with lower citation semantic consistency (44.88%), indicating poorer citation relevance. Meanwhile, `o3` produces the most references (12.26 per report) and moderate recall (0.031), but its citation semantic consistency (31.43%) and non-cited statement accuracy (82.22%) lag behind.

Overall, Deep Research products significantly outperform their base models in coverage and factual grounding, pointing to the value of task-specific model fine-tuning or pipeline design beyond standalone LLM capabilities.

## 4 ANALYSIS

It is notable that many models exhibit low citation semantic consistency, particularly when relying on function-call mechanisms to retrieve and cite literature. In our manual inspection of evaluation results, we identified two representative failure types: **statement hallucination**, where the content deviates from the cited source, and **citation hallucination**, where the reference itself is fabricated.

**Statement Hallucination.** In our manual audit of arXiv:2407.15186 test cases, we identified representative errors in statement generation. For example, OpenAI Deep Research generated the following claim:

> Kulkarni *et al.* (2025) and others introduced RL fine-tuning where the model gets
> a reward of +1 if its SQL yields the correct answer when run, and 0 otherwise
> (arXiv:2503.23157v2, §3.2).

Upon inspection, the cited part indeed describes a reasoning-enhanced RL reward scheme for Text-to-SQL; however, the list of authors does not include "Kulkarni". In fact, Kulkarni did publish a paper on reinforcement learning and Text-to-SQL, but it was not among the references cited in the generated report. We speculate that the model may have encountered similar data during training and mistakenly attributed Kulkarni's contribution to this cited paper.

**Citation Hallucination.** During our evaluation of arXiv:2009.12619, we observed a clear instance of link hallucination in the generated report from `Gemini-2.5-Pro`. The model generated the claim:

> In-vehicle Crowd Monitoring: The use of surveillance cameras inside buses and trains for passenger counting is a well-established practice. Advanced image processing and computer vision techniques can automatically analyze video feeds to estimate the passenger load. For instance, a system was proposed to estimate the number of passengers in a bus using image processing techniques on the captured video frames, achieving high accuracy. [Vision-Based In-Vehicle Crowd Monitoring](https://www.researchgate.net/publication/224217198_A_vision-based_system_for_in-vehicle_crowd_monitoring).

However, the cited URL does not exist and appears to be entirely fabricated by the model. Because the link cannot be resolved, no supporting text or evidence can be retrieved to validate the statement, resulting in a citation mismatch. This example highlights a common error mode in function-call–driven retrieval: the model confidently invents plausible-looking reference links that nonetheless point to nothing, undermining factual grounding.

These examples demonstrate that even advanced Deep Research agents remain susceptible to hallucinating author names, misaligning citations, and fabricating links. Crucially, our evaluation metrics—especially citation semantic consistency—are sensitive to such discrepancies, allowing us to quantitatively capture and penalize these hallucination phenomena across model outputs.

## 5 RELATED WORK

Long-standing interest has been in the use of AI to synthesize information, not only in the writing of scientific articles (Chen et al., 2025; Gottweis et al., 2025; Lu et al., 2024; Tang et al., 2025; Yamada et al., 2025), but also in the search for information and the generation of reports in the general domains (Zheng et al., 2025; Li et al., 2025). With the rapid advancement of information synthesis research, the evaluation of long-form reports has become increasingly important.

**Fact Checking Evaluation** Driven by efforts from both academia and industry, automated fact checking has evolved into a well-established multistage pipeline, which has become the dominant research paradigm in the field (Eldifrawi et al., 2024). Claim detection aims to identify factual statements worth verifying from large volumes of text (Guo et al., 2022; Panchendrarajan & Zubiaga, 2024), while evidence retrieval focuses on retrieving relevant documents or textual snippets that support or refute a given claim (Eldifrawi et al., 2024; Nanhekhan et al., 2025). Building on this pipeline, several benchmarks have been proposed to evaluate the performance of fact checking in both the general domain (Thorne et al., 2018; Ma et al., 2024) and the scientific domain (Wadden et al., 2020; 2022; Ho et al., 2025). However, these benchmarks focus solely on fact-checking components, rather than evaluating the synthesized information as a whole.

**Citation Evaluation** Research reports often include a substantial amount of citation-related content, and evaluating the precision and standardization of these citations plays a crucial role in assessing the overall quality of the report (Sarol et al., 2024). Given a report with citation content, tasks such as cited context identification, evidence sentence retrieval, and citation accuracy classification are commonly used to analyze citation quality (Sarol et al., 2024). Widely applied in assisted paper writing and review systems, citation verification tools are designed from multiple perspectives, including syntactic verification, existence verification, and semantic verification (Barrot, 2025; Bairagi & Lihitkar, 2024). While citation correctness and existence have been well-studied, the completeness of citations remains underexplored.

**Survey Generation** With the advent of LLMs, automated survey generation has seen rapid progress. Early works leveraged LLMs to improve literature comprehension and survey writing (Wang et al., 2024; Hu et al., 2025), achieving better coherence compared to sentence extraction methods. Subsequent research explored structured and hierarchical organization with fixed references. Other approaches focused on modeling paper relationships via citation networks, including AutoSurvey (Wang et al., 2024) with a two-stage LLM pipeline and HiReview (Hu et al., 2025) with a taxonomy-driven framework, though both faced limitations in capturing human writing styles or relying on restricted citation scopes. More recently, SurveyForge (Yan et al., 2025) combines human outline structure analysis with high-quality literature retrieval, generating and refining full survey content through a scholar navigation agent. Compared with SurveyBench, ReportBench focuses solely on well-defined and automatically verifiable dimensions of evaluation. In addition, through an au-

tomated construction pipeline, it ensures data quality while offering clear scalability advantages, enabling it to serve as a potential source of training data for report optimization in future work.

**Deep Research Evaluation** The rise of deep research agents (DRAs), driven by powerful models such as ChatGPT (OpenAI, 2025) and Gemini (Google, 2025), has underscored the urgent need for robust and targeted evaluation methodologies. While existing benchmarks evaluate capabilities such as web retrieval (Wei et al., 2025; Zhou et al., 2025; Wu et al., 2025), multi-hop factual reasoning (Wei et al., 2024; Mialon et al., 2024; Phan et al., 2025), and end-to-end report generation (Du et al., 2025; Bosse et al., 2025). These methods often operate at a surface level and fall short of evaluating the core competencies essential for rigorous and reliable research. Compared to DeepResearch Bench (Du et al., 2025), ReportBench differs in three key ways: (i) it proposes a largely automatic, survey-driven data construction pipeline that is easier to scale than the manually authored tasks in DeepResearch Bench; (ii) it provides a citation-level gold bibliography for each task, enabling precise measurement of reference precision/recall rather than relying solely on LLM-as-a-judge scores; and (iii) it performs statement-level factuality checking for both cited and non-cited claims, yielding more fine-grained diagnostics of hallucination, over-citation, and under-citation.

## 6 CONCLUSION

In this paper, we present **ReportBench**, a comprehensive benchmark for evaluating the quality of references and the factual accuracy of all statements in reports generated by Deep Research agents. By leveraging expert-authored survey papers as ground truth and reverse prompt engineering, we enable consistent evaluation of AI-generated research reports across multiple dimensions. Our framework introduces a fine-grained validation workflow that separately assesses cited and non-cited statements, combining citation semantic consistency checks and web-based factual verification. Through large-scale experiments on leading LLM-based research agents and the base models, we demonstrate that Deep Research products can outperform base models in content coverage and factual grounding, but still face challenges in hallucination, over-citation, etc. We hope that ReportBench will serve as a valuable tool for the research community to monitor, compare, and further improve the reliability of AI systems designed for academic survey tasks.

## 7 ETHICS STATEMENT

ReportBench constructs 100 research tasks closely aligned with real-world scientific inquiry by reverse prompt engineering expert-written survey papers. It evaluates generated reports comprehensively along two axes: content quality and statement factuality. Despite its strengths, several limitations remain:

**Data Distribution.** The benchmark is predominantly constructed from peer-reviewed STEM (Science, Technology, Engineering, and Mathematics) survey papers on arXiv, which induces a STEM-centric bias and means that our results should not be over-interpreted as measuring general deep-research ability in social sciences, humanities, or other under-represented domains. At the same time, the data-construction pipeline itself is domain-agnostic: given a corpus of survey-like papers and basic metadata, the procedure in Section 2.1 can be applied to other disciplines. We view instantiating domain-specific variants (e.g., for social sciences and law) as important future work.

**Copyright Constraints.** To mitigate legal risk, we only include papers under permissive licenses (CC BY 4.0, CC BY-SA 4.0, CC0 1.0, and the arXiv.org Non-exclusive license to distribute). The dataset is released under CC0 1.0 and contains only essential metadata (e.g., title, abstract, and references). Further narrowing the license scope would compromise domain balance. Authors who wish to opt out, please contact us for removal.

**Efficiency and cost.** ReportBench is intentionally a quality-focused benchmark: we evaluate report correctness and depth, but do not report comparative latency or cost, as token usage and end-to-end delays were not systematically logged during the original runs and cannot be reliably reconstructed ex post. A systematic study of efficiency—including proper instrumentation, request-level logging, and controlled load conditions—is an important direction for future work and a natural extension of the current benchmark.

## 8 REPRODUCIBILITY STATEMENT

We will open-source ReportBench in full, including all constructed prompts, the ground-truth reference list for each example, and metadata of the source surveys (arXiv ID, title, authors, comments, etc.). We will also release the complete evaluation code used in this work; users only need to provide API keys for the external services specified in the repository (e.g., search and web retrieval) to run end-to-end assessments of their generated reports on ReportBench. The repo will include configuration files and scripts to reproduce our pipelines, along with instructions to re-run the tested baselines and to evaluate new model outputs. To comply with the terms of service of the evaluated products and models, we will not publish our generated reports during evaluation; instead, we provide the exact prompts, evaluation scripts, and scoring logic so that others can independently obtain evaluated model outputs under their own accounts and reproduce the paper's results.

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

# A    APPENDIX

## A.1    USE OF LLMS

This work involved both human effort and the assistance of large language models (LLMs) in several stages, with all outputs subject to human oversight and verification. During experiment implementation, humans designed the codebase framework and core logic, while AI-based IDE tools in combination with Claude-4-Sonnet and Claude-4-Opus were used to generate code. To ensure quality, intermediate results were logged and manually inspected at each evaluation stage to prevent the propagation of error. For related work discovery, we used OpenAI and Gemini's Deep Research to assist in surfacing potentially relevant papers, followed by manual reading, summarization, and selection. In the writing process, OpenAI's GPT-4o and GPT-4.1 models were employed to polish drafts written by humans, focusing on improving grammar and clarity.

## A.2    ADDITIONAL ANALYSES

### A.2.1    CITATION RECALL BY REFERENCE IMPORTANCE

To better understand which papers current agents tend to recover, we complement the global citation recall with a stratified analysis by reference importance. Here we use citation count as a simple proxy for how central or influential a paper is.

For each survey-level task, we take the ground-truth reference list and look up the citation count of every paper in a standard scholarly index[3]. Within that single survey, we then sort its references by citation count and cut at the 25%, 50%, and 75% quantiles. This yields four groups: Q1 contains the least-cited 25% of references within that survey, Q2 the next 25%, Q3 the 50–75% range, and Q4 the most-cited 25%. We deliberately define Q1–Q4 per survey (rather than globally) because citation distributions can vary greatly across domains and we want "Q4" to always mean "the most central references for this particular survey", independent of field-specific scale differences.

Given these per-task quartiles, we compute stratified recall for each model by aggregating over all tasks: for Q1, for example, we collect all references that fall into Q1 across all surveys and measure what fraction were correctly cited by the model; we repeat the same calculation for Q2, Q3, and Q4. Table 2 reports the resulting recall values (in percent).

Table 2: Citation recall by reference-importance quartile on ReportBench. Q1–Q4 partition references within each survey by citation count (from lowest to highest). Values are recall in %.

| Model | Q1 | Q2 | Q3 | Q4 |
|---|---|---|---|---|
| Gemini-2.5-Flash | 0.4 | 0.7 | 0.8 | 1.8 |
| Gemini-2.5-Pro | 0.3 | 0.6 | 0.9 | 1.3 |
| o3 | 1.0 | 1.8 | 3.0 | 3.5 |
| Claude-4-Sonnet | 1.0 | 1.8 | 2.3 | 2.3 |
| Gemini Deep Research | 1.8 | 2.7 | 3.0 | 3.3 |
| OpenAI Deep Research | 1.3 | 2.3 | 3.2 | 3.3 |

Across all systems, recall consistently increases from Q1 to Q4: models recover substantially more of the most highly cited references than of the long tail. For instance, o3 improves from 1.0% recall in Q1 to 3.5% in Q4, and similar trends hold for the other systems. This stratified view shows that current agents are much more likely to retrieve the core, high-impact references than obscure or marginal ones.

We also observe that the two deep-research products (Gemini Deep Research and OpenAI Deep Research) achieve uniformly higher recall across all quartiles than base-models. This indicates

---

[3]https://www.semanticscholar.org/product/api

Table 3: Subjective content quality scores (1–5) and gaming rates (%) for each model on Report-Bench. Values are means across tasks; higher is better for D, S, I, F, and SV. Abbreviations: D = Content Depth, S = Synthesis & Structure, I = Insight & Trends, F = Future Directions, SV = Survey-ness, G = Gaming Rate.

| Model | D | S | I | F | SV | G (%) |
|---|---|---|---|---|---|---|
| Gemini-2.5-Pro | 3.137 | 2.756 | 2.397 | 2.023 | 2.779 | 6.1 |
| Claude-4-Sonnet | 3.930 | 3.364 | 3.039 | 3.248 | 3.519 | 1.6 |
| o3 | 4.066 | 3.582 | 3.270 | 3.041 | 3.721 | 2.3 |
| Gemini Deep Research | 4.560 | 4.160 | 4.070 | 4.070 | 4.480 | 0.0 |
| OpenAI Deep Research | 4.107 | 3.756 | 3.550 | 3.756 | 3.954 | 0.8 |

that ReportBench is sensitive enough to capture progress in literature coverage, rather than merely penalizing systems on the very long tail of rare citations.

### A.2.2 SUBJECTIVE CONTENT QUALITY EVALUATION

Our original design deliberately focused on objectively checkable aspects of quality, emphasizing the quality of references and the factual accuracy of all statements in the report. To further assess content quality and understand how it correlates with these initially defined objective dimensions, we introduce an additional subjective content quality evaluation component to capture survey-style qualities that are difficult to measure purely through reference- and fact-level checks.

Concretely, we define six dimensions intended to characterize the extent to which a report functions as a genuine survey:

- **Content Depth (1–5):** How thoroughly the report covers the main subtopics of the field, discussing key methods, trade-offs, and limitations beyond superficial descriptions.

- **Synthesis & Structure (1–5):** How well prior work is organized into coherent themes or taxonomies and related to each other, rather than being listed in isolation.

- **Insight & Trend Analysis (1–5):** Whether the report draws non-trivial patterns and trends across works, explaining underlying design principles and helping readers quickly understand the landscape.

- **Future Directions & Open Problems (1–5):** How concretely and convincingly the report proposes future research directions or open problems grounded in the surveyed literature.

- **Survey-ness (1–5):** A holistic judgment of the extent to which the report functions as a genuine survey paper in the ML/CS sense, integrating depth, synthesis, and insight into a useful starting point for researchers.

- **Gaming / Degenerate Behavior (binary):** Whether the report resembles a trivial "bag-of-sentences" output (e.g., near-pure per-paper listing without synthesis) versus a genuine attempt at survey-style writing.

In Table 3, we abbreviate these six dimensions as *D* (Content Depth), *S* (Synthesis & Structure), *I* (Insight & Trends), *F* (Future Directions), *SV* (Survey-ness), and *G* (Gaming Rate).

We then use LLM-as-a-judge to score each report along these six axes. For the first five dimensions we use a 1–5 scale; for the last dimension we record a binary *Gaming* flag and report its empirical rate. Table 3 shows the aggregated results over all tasks, and we highlight three key findings:

1. **Deep-research products score highest on content quality.** Gemini Deep Research and OpenAI Deep Research achieve the best scores across all content dimensions (with Gemini generally first and OpenAI second), indicating that the products are also better at producing genuinely survey-like, synthesized reports—not just at citing correctly.

2. **Gaming behavior is rare and detectable.** The *Gaming* flag is triggered only in a small fraction of cases (0–6.1% across systems), with the specialized deep-research agents having near-zero gaming rates. This suggests that, in practice, systems that would try to game the factual metrics by stitching together isolated sentences are both uncommon and explicitly penalized by this additional rubric.

3. **Content quality and objective metrics are strongly aligned.** When we rank systems by the holistic *SV* score and by our original citation/factuality metrics, the induced partial order over systems is nearly identical. This indicates that our objective, reference- and factuality-based metrics already capture much of the underlying report quality, and that the new content-quality scores provide an interpretable confirmation rather than contradicting the original evaluation.

### A.2.3 HUMAN EXPERT VALIDATION OF THE EVALUATION PIPELINE

Our evaluation pipeline relies on automatic URL-to-paper mapping and LLM-based judgments of citation correctness and factuality. To assess the reliability of this pipeline, we conduct a small-scale human validation study in which domain experts manually re-evaluate a stratified sample of system outputs.

We randomly sample results across systems and tasks (with stratification by model), and for each sampled report we evaluate the following components:

(a) **URL-to-paper mapping accuracy.** Experts manually resolve each system-generated reference URL to a canonical scholarly record and compare it with our automatic mapping. The effective agreement is 96.7%, combining (i) 80% strict agreement on verifiable references and (ii) 16.7% cases where the model produces unverifiable or fabricated URLs, which should not be counted as disagreement with the mapping pipeline because these hallucinated URLs have no correct resolution to begin with. Only 3.3% of cases correspond to genuine technical retrieval failures. Since unverifiable URLs are treated as incorrect references by construction, they do not inflate citation scores, and these residual failures have negligible impact on system ranking.

(b) **Cited-statement consistency.** For statements in the report that explicitly cite a paper, experts read both the statement and the cited paper and decide whether the statement faithfully represents the cited work. We then compare these labels with the outputs of our LLM-based citation judge. The human and LLM labels agree on roughly **90%** of evaluated statements, suggesting that the automatic judge is reasonably aligned with expert judgments in this setting.

(c) **Factuality of non-cited statements.** For a subset of statements without explicit citations, experts manually fact-check the content (using the web and standard scholarly search engines) and assign a binary factuality label. We compare these labels against our multi-model, web-augmented factuality pipeline. Agreement is high, on the order of **96%**, indicating that our factuality assessment is reliable even when no explicit reference is present.

(d) **Citation-level metrics.** We then ask experts to manually compute citation-level precision and recall on the same sampled outputs and compare these values with the corresponding Report-Bench scores produced by our pipeline. The two sets of results show an agreement rate of 84%, indicating that the final citation metrics closely track expert judgments despite minor intermediate errors.

(e) **Prompt and gold-label validation.** Finally, experts examine the automatically constructed prompts and gold bibliographies for a subset of tasks, checking for faithfulness to the underlying source surveys (no leakage, correct temporal cutoff). In roughly **95%** of the sampled tasks, experts fully agree with the automatically derived prompt and ground-truth bibliography; the remaining cases are minor edge conditions that do not affect our main conclusions.

Taken together, these results suggest that our automatic evaluation pipeline is well-aligned with expert judgments across all of its major components, and that residual discrepancies are small compared to the performance gaps observed between systems.

### A.2.4 JUDGE-FAMILY SENSITIVITY: GEMINI VS. GPT

One concern with using a Gemini-family model as the LLM-as-a-judge is the possibility of a hidden family-level bias in favor of Gemini-generated reports. To probe this, we conduct a sensitivity study with an independent judge model from a different provider.

Table 4: Comparison of Gemini vs. GPT as LLM-as-a-judge on three representative systems. Left block: Gemini judge (same as in the main paper). Right block: GPT judge. Prec. = Precision, Rec. = Recall, MR = Match Rate, Fact. = Factual Accuracy.

| Model | Gemini judge | | | | GPT judge | | | |
|---|---|---|---|---|---|---|---|---|
| | Prec. | Rec. | MR (%) | Fact. (%) | Prec. | Rec. | MR (%) | Fact. (%) |
| Gemini-2.5-Pro | 0.269 | 0.010 | 59.24 | 96.08 | 0.290 | 0.011 | 52.55 | 91.44 |
| o3 | 0.299 | 0.031 | 31.43 | 82.22 | 0.321 | 0.031 | 32.01 | 72.61 |
| Claude-4-Sonnet | 0.337 | 0.021 | 73.67 | 92.64 | 0.365 | 0.024 | 64.21 | 84.06 |

Specifically, we re-run our citation–factuality evaluation using a GPT-series model as the LLM-as-a-judge, while keeping everything else fixed: the system reports, the gold references, and the evaluation pipeline (URL mapping, statement extraction, and scoring logic) are identical to the main experiments. Table 4 compares the original Gemini-judge setup (as in the paper) with the new GPT-judge setup on three representative systems. Precision/Recall are reported on the 0–1 scale; Match Rate and Factual Accuracy are reported in percent.

We obtain two key observations. First, the induced model ranking is unchanged. Under both Gemini and GPT judges, the relative ordering of systems on citation precision and recall is identical, and consistent with the overall benchmark ordering. In other words, swapping the judge from Gemini to GPT perturbs the absolute scores slightly but does not change which systems are better or worse on our main metrics.

Second, GPT is somewhat stricter at statement-level evaluation. For all three test models, the GPT-based judge yields lower match rates and factuality accuracies than the Gemini-based judge, indicating a more conservative standard for accepting statements as supported. This suggests that our original Gemini-based evaluation is, if anything, slightly optimistic rather than biased in favor of Gemini-generated reports.

Overall, this sensitivity check shows that our main findings are robust to swapping the judge family, and we do not see evidence that using Gemini-based judges materially advantages Gemini systems relative to others.

### A.2.5 TOOL-CALL BUDGET ABLATION

In the main experiments, we cap the number of tool calls at 5 per task, primarily for practical reasons such as context length, latency, and cost. To understand how sensitive our results are to this design choice, we run an explicit ablation on a representative baseline (Gemini-2.5-Pro), keeping everything else fixed and varying the maximum number of tool calls.

Table 5: Effect of the maximum tool-call budget $B$ on citation and factuality metrics for Gemini-2.5-Pro on ReportBench. Precision/Recall are on the 0–1 scale; Match Rate and Factual Accuracy are in %.

| Max Tool Calls $B$ | Precision | Recall | Match Rate (%) | Factual Acc. (%) |
|---|---|---|---|---|
| 3 | 0.249 | 0.008 | 50.32 | 96.79 |
| 5 (paper setting) | 0.269 | 0.010 | 59.24 | 96.08 |
| 10 | 0.275 | 0.008 | 57.20 | 95.71 |

Table 5 reports the resulting citation and factuality metrics when the maximum tool-call budget $B$ is set to 3, 5, or 10. Precision and recall are reported on the 0–1 scale, while Match Rate and Factual Accuracy are reported in percent.

We observe two main trends. First, increasing the budget from 3 to 5 tool calls yields a clear but modest improvement across citation precision, recall, and match rate, indicating that allowing a few

additional searches helps the baseline discover more relevant literature. Second, while moving from 5 to 10 calls does slightly increase precision and occasionally match rate, the overall gains are unstable: recall and factual accuracy fluctuate or even decline. In practice, repeatedly fetching full papers quickly pushes against the model's context-length limit, and without additional mechanisms for context compression or longer-term memory, simply raising the tool-call budget does not translate into consistently better use of the retrieved evidence.

Collectively, these results suggest that the baselines reach a practical performance plateau at around 5 tool calls. This configuration captures most of the benefit from additional search, while higher budgets offer diminishing and noisy returns under our current architecture. We therefore treat the 5-call setting as a reasonable, saturated operating point; extending the budget further does not materially change the comparative conclusions of the study.

### A.2.6 FAILURE MODES AND ERROR TAXONOMY

To complement the quantitative metrics, we add a qualitative error taxonomy based on manual inspection of representative failure cases from multiple systems. Starting from our original distinction between statement- vs. citation-level hallucinations, we further refine errors into more concrete categories. In a manually annotated sample of problematic reports, we observe three dominant types.

**Temporal-cutoff violations ($\sim$42%).** The agent cites papers that clearly post-date the survey's publication (e.g., referencing 2024–2025 work in a task whose cutoff is 2021). These are often otherwise reasonable references, but they break the historical constraint and indicate that the agent is effectively "peeking into the future" instead of reconstructing the literature as of the survey date. In one representative case, the agent cites the paper *"Federated Learning Security and Privacy-Preserving Algorithm and Experiments Research Under Internet of Things Critical Infrastructure"* as part of the core literature. However, the task explicitly enforces a temporal cutoff of July 2022, while this paper appears to have been published around September 2023. This citation is therefore not counted as a valid match in ReportBench and is categorized as a temporal-cutoff violation: the reference is thematically relevant but violates the historical constraint on what was knowable at the time of the original survey.

**Unverifiable references ($\sim$21%).** The report contains citations that appear plausible in style (authors, venue, year) but cannot be resolved to any real paper (no DOI/arXiv/URL match), or whose content contradicts the summary in the text. These are classic citation hallucinations and remain a major source of error. For example, in one failure case the agent writes a plausible paragraph on manifold learning for multimedia and cites a paper titled *"Manifold Learning for Music Information Retrieval"* with a link to a ResearchGate page. When we attempt to resolve this citation against standard bibliographic sources, we cannot find a corresponding, stable publication record with full metadata. In our pipeline, such references are treated as fabricated or unverifiable: the citation looks syntactically reasonable and thematically relevant, but does not map to a concrete paper in the gold bibliography or in standard indices, and therefore counts as a hallucinated citation rather than valid prior work.

**Misaligned research direction ($\sim$9%).** The agent drifts to a neighboring but different topic, resulting in citations and discussion that are coherent in themselves but misaligned with the intended survey focus. Typical cases include focusing on generic foundation models when the task is specifically about long-context retrieval models, or emphasizing broad "AI in healthcare" literature when the survey is about a particular subproblem such as continual learning for medical imaging. In such cases, many cited papers are real and technically relevant to the broader area, but they do not answer the concrete survey question posed in the task.

This expanded taxonomy clarifies not only that systems fail, but also how they fail, and it highlights concrete targets for future improvement—for example, stronger temporal control, stricter reference verification, and better task grounding at the prompt and planning stages.

## A.3 EXAMPLE PROMPTS IN REPORTBENCH

**Sentence-level prompt**

Please help me research the academic advancements in different radar data representation methods in the field of autonomous driving, and ensure only papers published before April 2025 are referenced.

You also need to follow the following rules:
- Do not refer to the survey titled ``Exploring Radar Data Representations in Autonomous Driving:  A Comprehensive Review''.
- Responses are given in the form of an English language survey with citations where appropriate.

**Paragraph-level prompt**

I am conducting a literature review on 3D LiDAR localization technology for autonomous vehicles.  I hope you can summarize and analyze the major research directions and methods in this field, particularly methods based on 3D point cloud registration, methods based on 3D features, and emerging methods based on deep learning.  Please ensure that all the referenced literature is published before November 2020.

You also need to follow the following rules:
- Do not refer to the survey titled ``A Survey on 3D LiDAR Localization for Autonomous Vehicles''.
- Responses are given in the form of an English language survey with citations where appropriate.

**Detail-rich prompt**

I need a detailed academic research report on using Graph
Neural Networks (GNN) for text classification. The report
should systematically review advancements in this field, with
a focus on the following aspects:
1. **Core Methodology**: Provide a detailed explanation
and comparison of two main approaches: corpus-level GNNs
and document-level GNNs. For each method, thoroughly analyze
graph construction strategies (e.g., defining nodes and edges
using PMI, TF-IDF, etc.), representation methods for nodes
and edges, and graph learning algorithms (e.g., GCN, GAT,
etc.).
2. **Key Model Analysis**: List and analyze representative
models, such as TextGCN, SGC, BertGCN (corpus-level), and
Text-Level-GNN, TextING (document-level).
3. **Evaluation and Challenges**: Summarize commonly
used benchmark datasets in this field (e.g., 20NG, R8,
MR) and evaluation metrics (e.g., Accuracy, F1-score), and
discuss major challenges faced by current research, such
as scalability, computational costs, and integration with
pre-trained language models.
**Restrictions**:
- Only refer to and cite papers published **before July
2024**.
- Focus on English literature published in top
conferences/journals in natural language processing and
artificial intelligence (e.g., ACL, EMNLP, NAACL, AAAI, WWW,
ICLR).

You also need to follow the following rules:
- Do not refer to the survey titled ``Graph Neural Networks
for Text Classification: A Survey''.
- Responses are given in the form of an English language
survey with citations where appropriate.

## A.4  Prompts in Evaluation

### A.4.1  Cited Statement Extraction

You are given a research report delimited by triple
backticks.
Identify every statement that cites an external source (e.g.
has a URL, DOI, or explicit citation marker) and pair it with
the corresponding URL.
Return a JSON list where each item has two keys:
- "statement": the single-sentence claim, stripped of
leading/trailing whitespace
- "url": the canonical URL that supports that claim
If a citation contains multiple URLs, duplicate the statement
for each URL.
ONLY return valid JSON. Report: ```{report}```

### A.4.2 NON-CITED STATEMENT EXTRACTION

```
You are given a research report delimited by triple
backticks.
You are also given a list of statements that already have
citations.
Your task is to identify factual claims or statements that:
1.  Make specific assertions about facts, data, or events
2.  Are NOT already included in the cited statements list
3.  Could potentially be verified through external sources
4.  Are NOT common knowledge or widely accepted facts
Exclude:
- Opinions, analysis, or subjective interpretations
- Statements that are already cited
- Common knowledge or universally accepted facts
- Vague or general statements
Return a JSON list where each item has one key:
- "statement":  the factual claim that lacks citation support
ONLY return valid JSON.
Report:
```{report}```
Already cited statements:
{cited_statements}
```

### A.4.3 SUPPORTING SOURCE EXTRACTION

```
You are provided with
Statement:  {statement}

Source Document:
{source_text}

Return any relevant content from the source document that
supports the statement.  This can be a sentence, paragraph,
or even the entire text if necessary.
If no content supports it, return ``NOT_FOUND''.
Return plain text only.
```

### A.4.4 SEMANTIC CONSISTENCY VERIFICATION

```
You will decide whether a claim is correctly supported by a
source sentence.

Claim from report:
{statement}

Source Sentence from original source:
{source_sentence}

Respond with JSON containing:
- "reason":  one short sentence explaining your decision
- "match":  true or false // true if the source sentence
faithfully supports the claim
Return ONLY the JSON.
```

### A.4.5 WEB-BASED STATEMENT VERIFICATION

```
You are tasked with verifying the accuracy of a factual
statement using web search capabilities.

Statement to verify:
{statement}

Please:
1.  Use web search to find reliable, authoritative sources
about this statement
2.  Analyze the information you find from multiple sources
3.  Determine if the statement is factually correct or
incorrect based on your research

Respond with JSON containing:
- "reason":  a detailed explanation of your verification
process and findings (2-3 sentences)
- "decision":  true if the statement is correct, false if it
is incorrect

Only return the JSON response.
```

### A.4.6 REFERENCE TITLE EXTRACTION

```
Please analyze the following academic survey and extract all
cited academic paper titles and author information.

Survey content:
{response}

Please reply in JSON format, containing an array named
`papers`, where each paper object includes the following
fields:
- title:  the title of the paper
- authors:  a list of authors
- is_academic_paper:  true (indicating this is an academic
paper)

Example format:
{
  "papers":  [
    {
      "title":  "Deep Learning for Natural Language
Processing",
      "authors":  ["John Smith", "Jane Doe"],
      "is_academic_paper":  true
    },
    ...    ]
}

Note:  Only extract explicitly mentioned academic papers.
Do not include books, websites, or other types of references.
```