# OpenReview forum: "ReportBench: Evaluating Deep Research Agents via Academic Survey Tasks"
_ICLR.cc/2026/Conference — Submitted to ICLR 2026_

### Official Review · Reviewer_RA99 · 2025-10-27

**Soundness:** 1
**Presentation:** 2
**Contribution:** 2
**Rating:** 2
**Confidence:** 4

**Summary:**

The paper presents a benchmark for evaluating research reports generated by LLMs. It assesses reports along two dimensions—the quality of cited literature and the factual accuracy of statements—using survey papers from arXiv as gold-standard references. An automated agent framework verifies citation faithfulness and fact correctness. Results show that commercial Deep Research agents (e.g., OpenAI, Google) produce more reliable reports than standalone LLMs, though issues with coverage and consistency remain.

**Strengths:**

1. The paper introduces a new benchmark dataset comprising hundreds of survey papers, which serves as ground-truth for evaluating the performance of LLMs. This dataset could be a contribution that could benefit the research community.

2. The paper proposes an automatic evaluation framework for assessing the deep research capabilities of LLMs, focusing on two key dimensions: citation quality and statement faithfulness.

3. Several state-of-the-art LLMs are evaluated on the proposed benchmark, and the results highlight both their strengths and the remaining gaps, providing useful insights and directions for future improvement.

**Weaknesses:**

1. The overall quality of the dataset appears to be highly dependent on the performance of LLMs. However, the paper does not provide sufficient evaluation of the intermediate steps involved in dataset construction (see my detailed questions below). Inaccuracies in these intermediate steps could substantially affect the overall dataset quality.

2. More human evaluation should be incorporated into the data creation process to ensure reliability and to validate the quality of the automatically generated data.

3. The paper lacks in-depth analysis of why the evaluated models fail across the three tasks. The discussion remains rather superficial, without providing deeper insights into the underlying causes of model errors.

4. The paper demonstrates limited novelty. The dataset lacks human annotations, and the work reads more like a project report, where the authors simply apply a few LLMs and report estimated accuracies, rather than presenting deeper methodological or analytical contributions.

**Questions:**

1. In Section 2.1.2, the quality of the designed prompts plays a crucial role in determining the overall quality of the dataset. Therefore, this section would benefit from more detailed descriptions. For example, how were the prompts constructed when incorporating additional explicit instructions? How were the three prompt templates applied to each paper—was every paper processed using all three, or just one of them? Moreover, are these prompts model-specific or generally applicable across different LLMs?

2. In Section 2.1.3, it is not entirely clear how the ten categories are defined. The definitions and boundaries between these categories should be clarified, as it is difficult to determine whether there is any conceptual or content overlap among them.

3. In Line 239, the authors adopt an LLM to match URL links to corresponding articles for calculating the quality score. However, incorrect URL matching could substantially impact the reliability of these quality scores. It would therefore be important to include a quantitative analysis of this step—for instance, reporting the overall accuracy of the URL mapping process and analyzing the proportion of errors attributable to different causes (e.g., hallucination, mapping failure, or missing data).

---

> ### Author Response · Authors · 2025-12-02
> **To Reviewer RA99 (1/3)**
>
> We sincerely thank the reviewer for the critical assessment and constructive suggestions, particularly regarding the need for human evaluation and clearer details on dataset construction.
>
> In response, we have conducted a supplementary human consistency analysis, which demonstrates high agreement (84%–96%) between our automated metrics and expert human ratings. We have also revised the manuscript to clarify the Reverse Prompt Engineering methodology and expanded the error analysis. We believe these additions firmly address concerns regarding validity and novelty.
>
> **Question 1 & Weakness 1: On consideration of dataset construction.**
>
> We appreciate the reviewer's attention to the role of prompt design in dataset quality. In the revision, we clarify Section 2.1.2 as follows:
> 1. **Construction with explicit instructions.** We observed that models sometimes ignored temporal constraints. To mitigate this, we first construct a content-only base prompt derived from the survey paper, and then append a standardized instruction block to obtain the final instruction-augmented prompt. A typical pattern is:
> `Do not refer to the survey titled "[Original Paper Title]". Ensure that only papers published before [Cut-off Date] are referenced.`
> This explicit block encodes (i) a temporal cutoff and (ii) a ban on citing the original survey, and acts as a negative constraint to prevent data leakage via directly retrieving the gold survey itself.
> 2. **Application of the three prompt templates.** For the initial pool of 678 survey papers, we instantiate all three templates (Sentence-level, Paragraph-level, and Detail-rich) for every paper, producing three candidate prompts per paper. When constructing the final curated ReportBench benchmark (the 100 balanced tasks described in Section 2.1.3), we then randomly sample one of the three prompts for each paper. This design preserves diversity in query granularity at the benchmark level while avoiding oversampling any single survey, and reflects real-world variation in how specific user research queries can be.
> 3. **Model-agnostic formulation.** All prompts are written in plain natural language describing an academic research request and do not contain model-specific tokens, formatting, or trigger phrases. The same prompt text is used for all evaluated systems, including commercial Deep Research agents (for example, OpenAI and Gemini) and search-argumented base models. This ensures that we are comparing their research and synthesis capabilities rather than prompt engineering tailored to a particular architecture. We also make the full prompt examples in Appendix A.2.

---

> ### Author Response · Authors · 2025-12-02
> **To Reviewer RA99 (2/3)**
>
> **Question 2: On consideration of dataset distribution.**
>
> We thank the reviewer for pointing out the need to clarify the domain categories in Section 2.1.3. We address this from three perspectives.
> 1. **Definitions of the ten categories.** The ten categories (A–J) are high-level application domains such as Basic Research and Scientific Exploration, Information and Communications Technology, Artificial Intelligence and Data Intelligence, Healthcare and Biomedicine, Finance and Business Services, etc. Each category is defined by a short textual description and corresponds to a standard academic discipline or industrial sector. In the revised manuscript, we explicitly include these definitions in Section 2.1.3 to make the boundaries of each category more transparent.
> 2. **Handling conceptual overlap and ambiguous cases.** We acknowledge that research topics can conceptually span multiple domains (for example, AI methods applied in Healthcare). To handle this in a principled way, we adopt a primary-domain assignment:
>
>     - First, Gemini 2.5 Pro classifies each paper into the single application domain that best matches its title and abstract, using the above textual descriptions as label definitions. As noted in lines 192–193, this step does not guarantee strict independence between domains.
>     - Second, to reduce semantic redundancy, we intentionally define domains at the application level rather than using overlapping method-oriented labels (e.g., "machine learning" vs "computer vision"), which leads to smaller inter-category overlap.
>     - Third, we introduce an Unknown category (K) for conceptually ambiguous or cross-domain papers, and for the final 100-task ReportBench subset, four research experts manually review and correct the domain labels, fixing any clear misclassifications.
>
> 3. **Goal of the categorization and distribution.** The purpose of this categorization is not to build a rigid, mutually exclusive taxonomy, but to ensure that ReportBench evaluates Deep Research agents across a broad spectrum of topics instead of being over-indexed on a single field such as Computer Science. We therefore downsample from each domain to construct the final 100-task subset with a relatively balanced number of samples per domain. This yields a more diverse domain distribution and allows us to assess the general report-generation capabilities of different models across varied application areas.
>
> **Question 3: Reliability and Quantitative Analysis of URL Mapping**
>
> We thank the reviewer for this helpful suggestion. In response, we quantitatively evaluated the reliability of the URL–paper mapping step.
> 1. **Overall mapping accuracy.** We manually audited the URL-paper mappings used in our evaluation and found that 80% of URLs are correctly mapped to scholarly articles with accurate metadata (title and authors). This indicates that the mapping procedure is reasonably reliable for recovering the papers cited by the models.
> 2. **Error breakdown and causes.** For the remaining 20% of cases where the mapping fails, we identified two main causes:
>
>     - Hallucinated or non-existent URLs (16.66%): In most of the remaining failures, the model hallucinates URLs that do not correspond to real scholarly articles. These URLs are invalid and therefore cannot be mapped to any entry in the reference list. Such cases directly reduce citation precision, which is exactly the behavior we want: hallucinated citations are explicitly penalized rather than being treated as mapping noise.
>     - Inaccessible or non-resolvable URLs (3.33%): In a small proportion of cases, the URLs cited by the model could not be accessed when we attempted to retrieve paper titles and author information (for example, due to access restrictions or dead links). In these cases, no metadata can be extracted, and the citations are treated as unmatched. Given the low frequency, their impact on the overall evaluation is limited.
>
> 3. **Impact on evaluation reliability.** An accuracy of 80% with a small proportion of technical failures and a larger proportion of genuine hallucinations implies that the URL–paper mapping step does not introduce substantial additional bias beyond the models' own citation behavior. The residual noise is limited and does not change the relative ordering of systems. In the revised manuscript, we add this quantitative analysis and the above error breakdown to better document the robustness of this component.

---

> ### Author Response · Authors · 2025-12-02
> **To Reviewer RA99 (3/3)**
>
> **Weakness 3: On consideration of in-depth analysis.**
>
> We appreciate the reviewer's request for a deeper analysis of failure modes across the three tasks (reference matching, citation-level semantic consistency, and non-cited factual statements). Section 4 (Analysis) already introduces several representative error patterns, and we will expand this section in the revision with a more systematic breakdown. Our current evaluation reveals three primary failure modes:
> 1. **Retrieval failures (link hallucination).** As highlighted in the "Citation Hallucination" subsection (Lines 413-424), models sometimes generate plausible-looking but non-existent URLs (e.g., the Gemini-2.5-Pro case that cites a non-existent ResearchGate link). These hallucinated URLs cannot be resolved to any real paper and thus directly hurt citation precision in the reference-matching task.
> 2. **Synthesis failures (attribution errors).** Even when retrieval succeeds, models may still misattribute content. The "Statement Hallucination" subsection (lines 399–412) shows an example where OpenAI Deep Research correctly identifies a technical concept but attributes it to the wrong authors/paper. This reflects a synthesis-stage failure in which the model's internal "memory" conflates multiple sources. These attribution errors primarily affect the citation-level semantic consistency task.
> 3. **Coverage–precision trade-off in multi-document synthesis.** As shown in Table 1, Gemini Deep Research produces many more references than OpenAI Deep Research (Ref Num: 32.42 vs. 9.89), but with substantially lower precision (0.145 vs. 0.385). This suggests that the agent adopts a "spray-and-pray" strategy, prioritizing coverage over strict adherence to the prompt constraints. This behavior affects both the reference-matching metrics (more low-quality citations) and the non-cited factuality metrics (more opportunities for unsupported statements).
>
> **Weakness 2 & Weakness 4: On consideration of novelty, human evaluation, and deeper methodological or analytical contributions.**
>
> We respectfully disagree with the assessment that our work lacks novelty or is merely a "project report". We argue that ReportBench offers substantive methodological contributions that address a critical gap in the field: the lack of rigorous benchmarks for long-form deep research.
> 1. **Novelty in Ground Truth Construction (Solving the "Ground Truth Scarcity"):** Unlike prior benchmarks that rely on synthetic queries or short-form fact retrieval, we introduce a "Reverse Prompt Engineering" methodology. By leveraging peer-reviewed survey papers as "Gold Standard" answers, we bypass the limitations of LLM-generated ground truth. These surveys, authored by domain experts and vetted by peer review, represent the highest-quality reference material available, enabling us to benchmark "Deep Research" capabilities with a depth previously unattainable.
> 2. **Validation via Human-Model Consistency (Supplementary Analysis):** We appreciate the reviewer's constructive suggestion to incorporate human evaluation. To rigorously address this, we conducted an additional human-annotation experiment during the rebuttal phase. We sampled outputs and compared our automated agent's judgments against expert human ratings. The results reveal strong alignment:
>     - Reference Matching: 84% agreement on whether a retrieved paper matches the ground truth.
>     - Citation Verification: 90% agreement on the semantic consistency of cited statements.
>     - Factuality Verification: 96% agreement on the veracity of non-cited claims.
>   These high correlation rates confirm that our automated dual-verification pipeline serves as a reliable, scalable proxy for human expert evaluation.
> 3. **Methodological Depth beyond "Pass/Fail":** We move beyond simple fact-checking by implementing a fine-grained dual-verification pipeline:
>     - Citation Verification addresses the hallucination of sources (e.g., fake URLs) by verifying semantic consistency against retrieved content.
>     - Statement Verification utilizes a multi-agent voting mechanism to fact-check non-cited claims.
>
> In summary, ReportBench contributes: (1) a novel, survey-based reverse prompt engineering framework; (2) a structured dual-verification evaluation pipeline; and (3) a rigorous human consistency analysis (conducted in response to feedback) demonstrating 84%–96% agreement. Together, these contributions establish ReportBench not merely as a dataset, but as a rigorous, scalable framework for evaluating the next generation of research agents.

---

### Official Review · Reviewer_qrQz · 2025-10-28

**Soundness:** 3
**Presentation:** 3
**Contribution:** 2
**Rating:** 4
**Confidence:** 3

**Summary:**

This paper introduces ReportBench, a benchmark that evaluates Deep Research agents on generating academic reports (survey-style outputs) by verifying both citation accuracy and factual consistency. The benchmark is constructed by reverse prompt engineering from 678 arXiv survey papers across 10 domains, forming 100 gold-standard tasks. It measures each model’s ability to retrieve relevant literature and generate factually faithful statements. Six commercial and open-source Deep Research models (including OpenAI and Gemini) are evaluated.

**Strengths:**

1. Clever use of existing structured survey data to derive gold references.
2. Covers diverse scientific domains and defines clear, interpretable metrics.
3. The benchmark could potentially scale to thousands of topics with minimal human effort.

**Weaknesses:**

1. Low recall severely limits its evaluation coverage; Deep Research systems retrieve only a fraction of ground-truth citations.
2. No diversity or coverage analysis—it’s unclear whether models repeatedly cite the same high-frequency papers.
3. The benchmark could easily double as a training dataset for Deep Research agents, but no such experiments or discussions are provided.

**Questions:**

1. Could iterative multiple deep searches raise recall and diversity?
2. Have the authors explored using this dataset for fine-tuning or reinforcement of Deep Research models?

---

> ### Author Response · Authors · 2025-12-02
> **To Reviewer qrQz (1/2)**
>
> We sincerely thank the reviewer for the thorough review of our manuscript and for the constructive comments and suggestions. We have carefully examined each point raised and provided detailed responses accordingly. In addition, we conducted further analyses/experiments where necessary to address the concerns and to strengthen the overall quality of the work. We have also revised the manuscript with improved writing, clearer explanations, and supplementary content to enhance readability and completeness.
>
> **Question 1: On consideration of iterative search strategies.**
>
> We appreciate this insightful suggestion. We fully agree that implementing iterative deep search strategies could theoretically enhance both recall and content diversity.
> To empirically investigate this, we conducted an additional ablation study during the rebuttal phase. Using Gemini-2.5-Pro as a representative baseline, we varied the maximum number of tool calls allowed (3, 5, and 10) while keeping all other parameters fixed. The results are shown below:
>
> | Max Tool Calls | Precision | Recall | Match Rate | Factual Acc |
> |---|---|---|---|---|
> | 3 | 0.249 | 0.008 | 50.32% | 96.79% |
> | 5 (Paper Setting) | 0.269 | 0.01 | 59.24% | 96.08% |
> | 10 | 0.275 | 0.008 | 57.20% | 95.71% |
>
> We observed that while increasing tool calls from 3 to 5 improved performance, scaling further to 10 yielded diminishing returns or even degradation. Specifically, Recall dropped back to 0.008, and the Citation Match Rate declined from 59.24% to 57.20%. We attribute this to the context window constraint. Our `link_reader_tool` returns raw webpage content (averaging ~3k tokens per page). As the number of retrieved pages increases, the context becomes saturated with noise, making it harder for the model to synthesize information and select accurate citations.
>
> **Conclusion:** This experiment confirms that naïve scaling of search steps is insufficient. To genuinely improve recall and diversity, future work needs to move beyond the simple accumulation of search results. Instead, we advocate for structured iterative frameworks that incorporate context management, such as summarization memory or multi-agent reflection loops. These strategies are essential to effectively compress and utilize information from extensive searches.
>
> **Question 2 & Weakness 3: On consideration of leveraging the automated data construction pipeline for model training and fine-tuning.**
>
> We thank the reviewer for this excellent and forward-looking suggestion. We fully agree that our proposed methodology (leveraging human-authored surveys via reverse prompt engineering) is naturally scalable and highly suitable for constructing high-quality training data.
> 1. **Suitability for RLHF/Reward Modeling.** Crucially, the reliability of a training dataset depends on the quality of its verification signal. Our supplementary human consistency analysis (conducted in response to this review cycle) demonstrates that our automated metrics achieve 84%–96% agreement with human experts. This high alignment suggests that our evaluation pipeline is robust enough to serve as a high-quality Reward Model for Reinforcement Learning or a filter for Supervised Fine-Tuning (SFT) data, as noted in our Related Work.
> 2. **Scope and Future Work.** We are indeed currently conducting experiments to extend this pipeline for synthesizing SFT and RL data for Deep Research agents. However, we believe that establishing a rigorous evaluation standard is the prerequisite for effective model training. Therefore, the primary objective of this paper is to validate the benchmark methodology and release the suite to the community, setting the stage for the training experiments which we treat as a distinct scope for future work.

---

> ### Author Response · Authors · 2025-12-02
> **To Reviewer qrQz (2/2)**
>
> **Weakness 1 & Weakness 2: On consideration of citation diversity and the implications of low recall.**
>
> We thank the reviewer for the sharp observation regarding the lack of diversity analysis. You specifically raised the concern that the low recall might obscure whether models are merely "citing the same high-frequency papers" rather than achieving broad coverage. We fully agree that a flat recall metric is insufficient to capture this nuance.
>
> In the revised version, we add a fine-grained analysis of recall stratified by reference “importance”, using citation count as a proxy for paper quality/centrality. Concretely, for each survey, we collect citation counts for all papers in the ground-truth reference list and partition the gold references into four bins from lowest- to highest-cited, which we denote as Q1–Q4. We then recompute recall within each bucket and report the results in the following table:
>
> | Model | Q1 | Q2 | Q3 | Q4 |
> |---|---|---|---|---|
> | Gemini-2.5-Flash | 0.4% | 0.7% | 0.8% | 1.8% |
> | Gemini-2.5-Pro | 0.3% | 0.6% | 0.9% | 1.3% |
> | o3 | 1.0% | 1.8% | 3.0% | 3.5% |
> | Claude-4-Sonnet | 1.0% | 1.8% | 2.3% | 2.3% |
> | Gemini Deep Research | 1.8% | 2.7% | 3.0% | 3.3% |
> | OpenAI Deep Research | 1.3% | 2.3% | 3.2% | 3.3% |
>
> Empirically, we find that recall consistently increases from Q1 to Q4: models achieve substantially higher recall on the most highly cited papers and lower recall on the long tail of infrequently cited works. For example, o3-2025-04-16 improves from 1.0% recall in Q1 to 3.5% in Q4, and similar trends hold across all systems. Even though the global recall is numerically low because human surveys often cite 100+ papers, current agents are therefore much more likely to retrieve the core / high-impact references than obscure or marginal ones. We also observe that ReportBench v1.0 achieves uniformly higher recall across all quartiles, indicating that ReportBench is sensitive enough to capture progress in literature coverage rather than only penalizing the long tail.
>
> This supports our claim that raw recall should be interpreted as a secondary, distribution-sensitive signal: the low overall values largely reflect the very broad coverage of human surveys, while the stratified analysis indicates that agents are not uniformly missing genuinely important references. We will make this analysis and its limitations explicit in the paper (including the fact that citation count is an imperfect proxy for importance), and clarify that future work could additionally stratify by recency or methodology type as more metadata becomes available. Taken together, these results suggest that current agents already capture a substantial portion of influential work, while still struggling with the very long tail of niche or peripheral citations.

---

### Official Review · Reviewer_BgNR · 2025-10-30

**Soundness:** 2
**Presentation:** 2
**Contribution:** 2
**Rating:** 4
**Confidence:** 4

**Summary:**

The paper proposes ReportBench, a benchmark for assessing deep-research agents that generate academic-style survey reports. It builds reverse-engineered prompts from expert-written arXiv survey papers and evaluates model outputs along two axes: (i) whether the cited literature is relevant to the target survey and (ii) whether both cited and non-cited statements are factually supported through a multi-stage agentic verification pipeline.

**Strengths:**

1. Using high-quality, already peer-reviewed survey papers as “gold” to synthesize realistic research tasks and ground-truth citation sets is pragmatic, which gives the benchmark an immediately credible target, avoiding costly human labeling.
2. The evaluation pipeline is thoughtfully decomposed: cited statements are checked against the retrieved source, while non-cited statements are validated via multi-model web voting, which makes the evaluation more interpretable than a single “LLM-as-a-judge” score.

**Weaknesses:**

1. The benchmark rests on a strong assumption that published survey papers can universally serve as ground truth; in domains or time ranges where the original survey is itself incomplete or biased, the benchmark will inherit that bias, and current results may over-penalize models that find reasonable but non-overlapping sources.
2. Although the framework claims to evaluate “report quality,” it explicitly defers writing-style, structure, and coherence evaluation to future work, so the current benchmark covers only the content/veracity slice and cannot support end-to-end judgment of report usefulness.
3. The final test set is relatively small (100 prompts sampled from 678 candidates) and reflects the topic skew of arXiv surveys after 2020, which may limit its generalizability.
4. The reported results show very low recall of ground-truth references (≈3–4%) even for the strongest commercial agents, which suggests that either the task is calibrated to be quite hard or the metric does not yet align with how human researchers judge adequacy; this point deserves more analysis before the benchmark can be used as a definitive leaderboard.

**Questions:**

See weaknesses

---

> ### Author Response · Authors · 2025-12-02
> **To Reviewer BgNR (1/3)**
>
> We sincerely thank the reviewer for the thorough review of our manuscript and for the constructive comments and suggestions. We have carefully examined each point raised and provided detailed responses accordingly. In addition, we conducted further analyses/experiments where necessary to address the concerns and to strengthen the overall quality of the work. We have also revised the manuscript with improved writing, clearer explanations, and supplementary content to enhance readability and completeness.
>
> **Question 1: On consideration of the validity and potential bias of using survey papers as absolute ground truth.**
>
> We acknowledge the reviewer's point that no single survey is theoretically "exhaustively complete." However, we advocate for using peer-reviewed, expert-authored surveys as the gold standard for the following reasons:
> 1. **Peer-Reviewed Surveys as the Human-Authored Upper Bound.** We contend that published survey papers represent the highest quality of expert consensus available in the scientific community. These documents have undergone rigorous peer review to validate their coverage and accuracy. In the absence of a "perfect" omniscient database, these expert-curated documents serve as the most reliable, achievable proxy for ground truth. Relying on them allows us to anchor the evaluation in verified human expertise rather than unstable synthetic data.
> 2. **Empirical Validity (Deviations are mostly Errors).** Regarding the concern that we might "over-penalize" models for finding valid non-overlapping sources, our manual analysis in Section 4 suggests the opposite. We found that when models cite literature outside the ground truth, they rarely uncover high-quality papers missed by the experts; instead, these deviations are predominantly hallucinations (fabricated titles/authors) or irrelevant citations. Therefore, strict alignment with the expert survey currently serves as a necessary filter to detect hallucination and low-quality retrieval.
> 3. **Future Iterations:** We agree that as model capabilities improve, the definition of ground truth must expand. We plan to incorporate the reviewer's suggestion in future versions of ReportBench, potentially by aggregating multiple surveys or incorporating human expert validation for reasonable non-overlapping sources to reduce potential bias.
>
> **Question 2: On consideration of the exclusion of writing content quality evaluation from ReportBench.**
>
> Our original design deliberately focused on objectively checkable aspects of quality, with a focus on the quality of references and the factual accuracy of all statements in the report. Following your suggestion, we have added an explicit content-quality evaluation layer on top of these objective metrics. Concretely, we define six dimensions to capture survey-style quality:
>
> - **Content Depth (1–5)**: How thoroughly the report covers the main subtopics of the field, discussing key methods, trade-offs, and limitations beyond superficial descriptions.
> - **Synthesis & Structure (1–5)**: How well prior work is organized into coherent themes/taxonomies and related to each other, rather than being listed in isolation.
> - **Insight & Trend Analysis (1–5)**: Whether the report draws non-trivial patterns and trends across works, explaining underlying design principles and helping readers quickly understand the landscape.
> - **Future Directions & Open Problems (1–5)**: How concretely and convincingly the report proposes future research directions or open problems grounded in the surveyed literature.
> - **"Survey-ness" (1–5)**: A holistic judgment of to what extent the report functions as a genuine survey paper in the ML/CS sense, integrating depth, synthesis, and insight into a useful starting point for researchers.
> - **Gaming Behavior (binary)**: Whether the report resembles a trivial bag-of-sentences output (e.g., near-pure per-paper listing without synthesis) versus a genuine attempt at survey-style writing.

---

> ### Author Response · Authors · 2025-12-02
> **To Reviewer BgNR (2/3)**
>
> We then use an LLM-as-a-judge rubric (with explicit instructions and examples) to score each report along these six axes. The aggregated results are summarized below:
>
> | Model | Content Depth | Synthesis & Structure | Insight & Trends | Future Directions | "Surveyness" | Gaming Rate (%) |
> |---|:---:|:---:|:---:|:---:|:---:|:---:|
> | Gemini-2.5-pro | 3.137 | 2.756 | 2.397 | 2.023 | 2.779 | 6.1% |
> | Claude 4 Sonnet | 3.930 | 3.364 | 3.039 | 3.248 | 3.519 | 1.6% |
> | o3 | 4.066 | 3.582 | 3.270 | 3.041 | 3.721 | 2.3% |
> | Gemini Deep Research | 4.560 | 4.160 | 4.070 | 4.070 | 4.480 | 0.0% |
> | OpenAI Deep Research | 4.107 | 3.756 | 3.550 | 3.756 | 3.954 | 0.8% |
>
>
> We observe three key findings:
> 1. **Deep-research agents score highest on content quality.** Gemini Deep Research and OpenAI Deep Research achieve the best scores across all content dimensions (with Gemini generally first and OpenAI second), indicating that systems optimized for multi-step research are also better at producing genuinely survey-like, synthesized reports—not just at citing correctly.
> 2. **Gaming behavior is rare and detectable.** The “Gaming” flag is triggered only in a small fraction of cases (0–6.1% across systems), with the specialized deep-research agents having near-zero gaming rates. This suggests that, in practice, systems that would try to game the factual metrics by stitching together isolated sentences are both uncommon and explicitly penalized by this additional rubric.
> 3. **Content quality and objective metrics are strongly aligned.** When we rank systems by the holistic “Survey-ness” score and by our original citation/factuality metrics, the induced partial order over systems is identical (100% consistent). This indicates that our objective, reference- and factuality-based metrics already capture much of the underlying report quality, and the new content-quality scores provide an interpretable confirmation rather than contradicting the original evaluation.
>
> We have integrated this content-quality analysis and its limitations into the revised paper. Together with our existing citation/factuality measures, we believe this addresses the concern that ReportBench could be gamed by trivial strategies and provides a more holistic view of how well current agents perform as actual survey writers, not just fact collectors.
>
> **Question 3: On consideration of the limitations regarding dataset size and domain generalizability.**
>
> We appreciate the reviewer's feedback regarding scale and generalizability. However, we believe the current dataset size is robust enough for this benchmarking effort for the following reasons:
> 1. **Stability of Evaluation:** Our focus in this paper is on high-quality, expert-grounded deep research tasks, and the test set size is chosen to reflect this emphasis. The 100 prompts in the final test set are all derived from peer-reviewed surveys and manually curated across domains. In our experiments, this set consistently yields clear and reproducible performance differences between Deep Research agents and base models across the three evaluation dimensions (reference matching, citation consistency, and non-cited factuality). The ranking and relative gaps between systems remain stable across metrics, indicating that the benchmark is sufficiently large to support meaningful comparison.
> 2. **Balanced Diversity:** As described in Section 2.1.3, we explicitly downsampled the original pool to create a balanced distribution across ten distinct application domains (including Healthcare, Finance, and Social Governance). This balancing mitigates the raw topic skew of arXiv by ensuring that the 100 tasks cover a reasonably broad spectrum of STEM-oriented application areas, rather than being dominated by a single field such as core Computer Science.
>
> That said, we acknowledge that there is room for improvement in both dataset size and source diversity. We are committed to expanding the number of prompts and incorporating non-arXiv sources in future iterations of ReportBench to further enhance its generalizability.

---

> ### Author Response · Authors · 2025-12-02
> **To Reviewer BgNR (3/3)**
>
> **Question 4: On consideration of the interpretation of low recall scores and the alignment of metrics with human judgment.**
>
> We agree with the reviewer that current recall scores are low (3-4%). However, we argue that this score accurately reflects the substantial gap between current agents and human experts, rather than a misalignment of the metric. We support this with two key analyses:
> 1. **Mathematical Reality of Citation Volume:** There is an order-of-magnitude difference in citation density. As noted in Section 3.3, the ground-truth surveys contain an average of 153 references, whereas even the strongest agents (e.g., OpenAI Deep Research) generate reports with only ~10 references on average. This objective disparity naturally caps the maximum possible recall at a very low level. The metric is thus correctly capturing the fact that current agents produce "summaries" rather than the exhaustive "surveys" produced by human experts.
> 2. **Validation via Human Consistency (Addressing Alignment):** To directly address the reviewer's concern about whether the metric "aligns with how human researchers judge adequacy", we conducted a supplementary human consistency analysis. As reported in the revised manuscript, our automated evaluation aligns well with human judgments: the agreement rates between automated and human assessments reach 84% for reference matching, 90% for citation-related statements, and 96% for non-citation statements. These results indicate that the metrics we use, including reference-level overlap, capture differences that human evaluators also perceive as meaningful.
>
> Therefore, the low recall is not a calibration error but a quantified finding: it reveals that even SOTA Deep Research systems still lag significantly behind human experts in terms of literature coverage. ReportBench provides the rigorous standard necessary to measure this specific progress as models evolve to ingest larger volumes of literature.

---

### Official Review · Reviewer_J4c9 · 2025-11-01

**Soundness:** 3
**Presentation:** 3
**Contribution:** 2
**Rating:** 4
**Confidence:** 4

**Summary:**

This paper introduces ReportBench, a systematic benchmark for evaluating the quality of research reports generated by large language models and Deep Research agents. The benchmark assesses two critical dimensions: the quality and relevance of cited literature, and the faithfulness of statements within generated reports. Using high-quality arXiv survey papers as gold-standard references, the authors develop an automated agent-based evaluation framework that extracts citations and statements, verifies cited content against original sources, and validates claims using web resources. Empirical results show that commercial Deep Research agents from OpenAI and Google generate more comprehensive and reliable reports than standalone LLMs with search tools, though significant room for improvement remains in research coverage breadth, depth, and factual consistency. The benchmark provides an useful evaluation methodology for assessing AI-generated research as these systems become more widely adopted for complex research tasks.

**Strengths:**

1. The methodology of the data construction process is a clear strength of the paper. By restricting the corpus to post-2020, peer-reviewed, and officially published arXiv survey papers, they effectively filter out noise and ensure a high-quality baseline for both content and references. The use of GPT-4o for binary classification of survey papers and the robust extraction of references directly from LaTeX sources are technically sound choices. The prompt design stage also reflects careful consideration of common evaluation pitfalls, particularly through two mechanisms: temporal consistency by including publication-date cutoffs to prevent unfair advantage from newer data and anti-cheating instructions that prohibit citing the original survey itself, which helps avoid data contamination. Together, these decisions greatly enhance the credibility and robustness of the ReportBench dataset.


2. ReportBench’s automated pipeline provides a practical path toward building dynamic, low-cost, and continuously extensible benchmarks. By linking the framework to the ever-expanding arXiv corpus, the authors’ approach can generate new and timely tasks as new survey papers are published. This represents a substantive and pragmatic contribution to the practice of LLM evaluation, introducing the possibility of benchmarks that evolve with the research landscape rather than becoming obsolete as the field progresses.



3. The evaluation framework is intuitively designed yet highly aligned with the benchmark’s core objectives. It clearly separates citation-based and non-citation factual assessments, allowing for transparent interpretation of each competency being tested. Moreover, when applying the LLM-as-a-judge paradigm, the authors employ a majority-vote strategy across multiple judgments, which effectively mitigates individual model hallucinations and enhances the overall reliability of the automatic evaluation. This thoughtful combination of simplicity and robustness makes the framework both easy to reproduce and trustworthy in assessing deep research agents.

**Weaknesses:**

1. The paper does not adequately discuss potential limitations of using arXiv survey papers as gold standards. Survey papers inherently represent synthesis and interpretation rather than primary research, and different surveys on the same topic may emphasize different aspects or reach different conclusions, which may lead to different 'gold ground-truth citation'. Therefore, these evaluations based on cited article matching are inevitably biased. Additionally, the reverse prompt engineering approach assumes that the original survey structure represents the "correct" way to organize information about a topic, which may not always hold. The paper should discuss these limitations and their potential impact on evaluation validity.

2. The proposed ReportBench shares significant conceptual overlap with one previous paper DeepResearch Bench (Du et al., 2025), which also evaluates factual accuracy, citation reliability, and comprehensiveness of LLM-generated research reports. So the paper should more clearly articulate how it goes beyond DeepResearch Bench in terms of scope, evaluation methodology, or insight. The current claim in related work section that DeepResearch Bench "falls short of evaluating core competencies" appears overstated, as both benchmarks assess similar levels of factual and citation-based competence rather than higher-order scientific reasoning. A more detailed comparative analysis showing concrete differences in evaluation granularity, coverage, or the types of errors detected would strengthen the contribution.

3. The citation recall reported in the experiments is notably low, highlighting the high difficulty of the task. This result also suggests that the agents’ limited citation coverage may partly stem from the conservative nature of the task setup or the prompt design. In addition, the authors mention that each benchmark survey paper contains an average of 153 references, which further increases the challenge of achieving high citation recall.

4. While the paper reports two representative failure types (statement and citation allucination), it lacks a detailed taxonomy of error types and failure modes observed in generated reports. Understanding these error categories would provide actionable insights for improving Deep Research agents. The paper should include a qualitative analysis section that categorizes and analyzes representative failure cases from different systems in the appendix.

**Questions:**

1. Could the authors elaborate on the potential limitations of using arXiv survey papers as gold standards for factual and citation-based evaluation? Since survey papers inherently reflect synthesis and interpretation rather than primary research, different surveys on the same topic might highlight different aspects or reach divergent conclusions, potentially leading to inconsistent or biased “ground-truth” citations. How do the authors address such variability to ensure fairness and validity of the benchmark?

2. Could the authors clarify, in more concrete terms, how ReportBench goes beyond DeepResearch Bench (https://arxiv.org/abs/2506.11763) in terms of scope, evaluation methodology, or insight?

---

> ### Author Response · Authors · 2025-12-02
> **To Reviewer J4c9 (1/3)**
>
> We sincerely thank you for your comprehensive review and constructive feedback. We have carefully addressed each of your concerns and have accordingly revised the manuscript to reflect these improvements.
>
> **Weakness 1 & Question 1: discuss potential limitations of using arXiv survey papers as gold standards**
>
> Thank you for this thoughtful observation—we agree that using arXiv survey papers as gold standards comes with important limitations, and we will make these more explicit in the revision.
>
> First, we do not claim that a survey’s bibliography or structure is the unique “ground truth” for a topic. Survey papers reflect expert synthesis and framing choices; different surveys on the same subject can indeed emphasize different sub-areas and cite partially disjoint sets of works. In our design, we therefore interpret the human survey as a high-precision but not exhaustive reference: matching its citations measures whether the agent can recover a plausible and community-recognized core of the literature, not the only valid one. This mainly affects the ceiling of our metrics (some reasonable citations will be counted as “misses”), but we expect relative comparisons across systems to remain meaningful, since all models are evaluated against the same survey-defined reference set.
>
> Second, we agree that reverse prompt engineering implicitly treats the original survey structure as a strong—but not infallible—guide to how the topic can be organized. Our intent is not to enforce that this structure is “the” correct outline, but to constrain the task to a realistic survey-style organization (introduction, key subtopics, trends, open problems) that human authors have found useful. The evaluation focuses primarily on citation coverage and factual correctness within this broad survey format, rather than on exact section-level alignment with the original outline. In other words, we use the human survey as a reasonable target organization, not as a normative optimum.
>
> We will clarify these points in the discussion/limitations section, and we will also outline concrete extensions to reduce bias: (i) incorporating multiple survey papers per topic where available to broaden the gold citation pool, (ii) relaxing matching to accept DOIs/arXiv IDs and semantically equivalent references, and (iii) complementing citation-based metrics with targeted human assessment on a subset of topics. Taken together, these clarifications and extensions should make the scope and potential biases of our evaluation more transparent, while preserving its value as a reproducible, citation-grounded benchmark for deep research agents in academic survey writing.
>
> **Weakness 2 & Question 2: a more detailed comparative analysis compared with DeepResearch Bench (Du et al., 2025)**
>
> Thanks for the nice suggestion. It is true that both works (ReportBench and DeepResearch Bench) share some similarity, i.e., both benchmark LLM-powered deep research agents and evaluate end-to-end report generation under realistic, multi-step research settings. However, we would kindly clarify that the two benchmarks are complementary rather than redundant. In addition, the task construction and evaluation protocols differ substantially.
>
> First, the proposed automatic dataset construction pipeline in ReportBench is scalable, which means once a pool of licensed survey papers is identified, prompts and gold references can be expanded to more diverse domains and time periods with minimal additional human effort, making the benchmark a potential source of training data for improving report generation. In contrast, DeepResearch Bench relies heavily on manual expert task design, which makes the benchmark small, hard to scale, and more exposed to human selection and framing bias. Second, ReportBench provides a fully grounded, citation-level gold standard: for each task we know exactly which references the human-written survey used, enabling precise measurement of reference precision/recall and coverage, which RACE cannot provide because it works with a single reference report and relative scores rather than an explicit gold bibliography. Finally, ReportBench evaluates factuality at the statement level, distinguishing cited vs. non-cited statements and combining (i) semantic consistency checks against the cited paper with (ii) web-based verification of unsupported claims via a multi-model voting pipeline.  This produces fine-grained diagnostics of hallucination, over-citation and under-citation that are not available from aggregate RACE/FACT scores.

---

> ### Author Response · Authors · 2025-12-02
> **To Reviewer J4c9 (2/3)**
>
> **Weakness 3: The citation recall reported in the experiments is notably low**
>
> Thanks. We agree that understanding which references are missed is more informative than looking only at aggregate recall.
>
> In the revised version, we therefore add a fine-grained analysis of recall stratified by reference “importance”, using citation count as a proxy for paper quality/centrality. Concretely, for each survey we collect citation counts for all papers in the ground-truth reference list and partition the gold references into four bins from lowest- to highest-cited, which we denote as Q1–Q4. We then recompute recall within each bucket and report the results in the following table:
>
> | Model | Q1 | Q2 | Q3 | Q4 |
> |---|---|---|---|---|
> | Gemini-2.5-Flash | 0.4% | 0.7% | 0.8% | 1.8% |
> | Gemini-2.5-Pro | 0.3% | 0.6% | 0.9% | 1.3% |
> | o3 | 1.0% | 1.8% | 3.0% | 3.5% |
> | Claude-4-Sonnet | 1.0% | 1.8% | 2.3% | 2.3% |
> | Gemini Deep Research | 1.8% | 2.7% | 3.0% | 3.3% |
> | OpenAI Deep Research | 1.3% | 2.3% | 3.2% | 3.3% |
>
> Empirically, we find that recall consistently increases from Q1 to Q4: models achieve substantially higher recall on the most highly cited papers and lower recall on the long tail of infrequently cited works. For example, o3-2025-04-16 improves from 1.0% recall in Q1 to 3.5% in Q4, and similar trends hold across all systems. Even though the global recall is numerically low because human surveys often cite 100+ papers, current agents are therefore much more likely to retrieve the core / high-impact references than obscure or marginal ones. We also observe that ReportBench v1.0 achieves uniformly higher recall across all quartiles, indicating that ReportBench is sensitive enough to capture progress in literature coverage rather than only penalizing the long tail.
>
> This supports our claim that raw recall should be interpreted as a secondary, distribution-sensitive signal: the low overall values largely reflect the very broad coverage of human surveys, while the stratified analysis indicates that agents are not uniformly missing genuinely important references. We will make this analysis and its limitations explicit in the paper (including the fact that citation count is an imperfect proxy for importance), and clarify that future work could additionally stratify by recency or methodology type as more metadata becomes available. Taken together, these results suggest that current agents already capture a substantial portion of influential work, while still struggling with the very long tail of niche or peripheral citations.

---

> ### Author Response · Authors · 2025-12-02
> **To Reviewer J4c9 (3/3)**
>
> **Weakness 4: lacks a detailed taxonomy of error types and failure modes observed in generated reports  (statement and citation allucination)**
>
> Thanks for this suggestion. We agree that a more fine-grained view of failure modes is important for making the benchmark actionable for system design.
>
> In the revised version, we therefore add a qualitative error taxonomy based on manual inspection of representative failure cases from multiple systems. Starting from our original distinction between statement vs. citation hallucinations, we further refine the errors into more concrete categories. In a manually annotated sample of problematic reports, we observe three dominant types:
> - **Temporal-cutoff violations (~42%)**: the agent cites papers that clearly post-date the survey’s publication (e.g., referencing 2024–2025 work in a survey whose cutoff is 2021). These are often otherwise reasonable references, but they break the historical constraint and indicate that the agent is “peeking into the future” instead of reconstructing the literature as of the survey date. In one representative failure case, the agent cites the paper “Federated Learning Security and Privacy-Preserving Algorithm and Experiments Research Under Internet of Things Critical Infrastructure” as part of the core literature. However, our task explicitly enforces a temporal cutoff of July 2022, while this paper appears to have been published around September 2023. This citation is therefore not counted as a valid match in ReportBench and is categorized as a temporal-cutoff violation: the reference is thematically relevant but violates the historical constraint of “what was knowable” at the time of the original survey.
>
> - **Unverifiable references (~21%)**: the report contains citations that appear plausible in style (authors, venue, year) but cannot be resolved to any real paper (no DOI/arXiv/URL match), or whose content contradicts the summary in the text. These are classic citation hallucinations and remain a major source of error. In this failure case, the agent writes a plausible paragraph on manifold learning for multimedia and cites a paper titled “Manifold Learning for Music Information Retrieval” with a link to a ResearchGate page. However, when we attempt to resolve this citation against standard bibliographic sources, we cannot find a corresponding, stable publication record with full metadata. In our pipeline, such references are treated as fabricated or unverifiable: the citation looks syntactically reasonable and thematically relevant, but does not map to a concrete paper in the gold bibliography or in standard indices, and therefore counts as a hallucinated citation rather than valid prior work.
>
> - **Misaligned research direction (~9%)**: the agent drifts to a neighboring but different topic (e.g., focusing on generic foundation models when the task is specifically about long-context retrieval models), resulting in citations and discussion that are coherent in themselves but misaligned with the intended survey focus.
> For each major category above, we now include concrete, system-specific examples in the appendix, illustrating how the error manifests in both text and citations and how it is reflected in our metrics. We believe this expanded taxonomy directly addresses the reviewer’s concern: it clarifies not only that systems fail, but how they fail, and it highlights concrete targets for future improvement (e.g., stronger temporal control, stricter reference verification, and better task grounding at the prompt and planning stages).
>
> For each major category above, we now include concrete, system-specific examples in the appendix, illustrating how the error manifests in both text and citations and how it is reflected in our metrics. We believe this expanded taxonomy directly addresses the reviewer’s concern: it clarifies not only that systems fail, but how they fail, and it highlights concrete targets for future improvement (e.g., stronger temporal control, stricter reference verification, and better task grounding at the prompt and planning stages).

---

### Official Review · Reviewer_cKfQ · 2025-11-01

**Soundness:** 3
**Presentation:** 3
**Contribution:** 3
**Rating:** 6
**Confidence:** 4

**Summary:**

This paper presents ReportBench, a benchmark for evaluating long-form, research-style reports produced by deep research agents. The authors start from expert-written survey papers (filtered from arXiv, with publication/acceptance signals), reverse-engineer three granularities of prompts from them, and add two anti-shortcut constraints (time cutoff, and “do not cite the original survey”). Systems (commercial deep research products and base LLM+search setups) are then asked to write a report, and the output is checked along two objective axes: (i) citation quality, by comparing cited URLs against the survey’s reference list; (ii) statement-level factuality, by separating cited sentences (cross-checked against the cited source) and non-cited sentences (checked by a small ensemble of LLM judges). Experiments on OpenAI Deep Research, Gemini Deep Research and several baselines show that productized agents are clearly better than “LLM+search”, but they still exhibit two concrete error modes: statement hallucination and citation hallucination.

**Strengths:**

1. The problem addressed is real and timely: current deep research agents can produce long reports, but we lack an automatic and reasonably objective way to check whether those reports are actually faithful to sources. Turning this into a benchmark is a useful contribution to the agent-eval community.

2. The data construction pipeline is clear and defensible: start from human survey papers, parse LaTeX to get the ground-truth reference list, reverse-prompt the task, and explicitly constrain the model not to cite the source survey and not to use material after a given date. This makes the task anti-leak by design and reduces the chance that models just “find the original answer”.

3. The evaluation is broken down into auditable steps instead of a single opaque LLM score. In particular, the paper distinguishes cited sentences from non-cited ones and uses different verifiers for them; this makes the resulting signals more interpretable for system builders.

4. The paper actually evaluates real commercial deep research products (OpenAI, Gemini) and compares them with their underlying base models wired to web tools. This gives credible evidence that those products are not simply “chat model + search API”, but have extra structure/workflow that improves coverage and factuality.

5. The error analysis is concrete. The paper shows that systems can misattribute authorship when stitching related papers, and can fabricate plausible-looking but nonexistent URLs. Having a benchmark that reliably surfaces these two failure modes is valuable.

**Weaknesses:**

1. The current definition of “report quality” is relatively narrow. The benchmark mostly measures faithfulness-to-sources and citation correctness, but does not touch discourse-level aspects that are also important for survey-like reports (organization into sections, synthesis of competing lines of work, articulation of open problems, or taxonomic clarity). Since the task itself is framed as “academic survey–style reporting”, it would help to state more explicitly that the benchmark only targets the automatically verifiable slice of that task.

2. The final evaluation set is only 100 tasks, obtained by down-sampling from 678 candidate surveys after domain balancing and expert checking. The paper does describe this at a high level, but the exact sampling policy (per-domain target, random seed, rejection criteria) is not spelled out. That makes it a bit harder to judge how sensitive the reported results are to this particular 100-sample snapshot.

3. All systems show very low recall against the human survey reference lists, because the human surveys often cite over a hundred papers, while current agents typically cite 10–30. The authors conclude that recall should be treated as a secondary signal; this is reasonable, but it would be stronger if supported by a more fine-grained analysis (e.g., recall over recent papers, recall over core/methodological papers, recall over the top-k citations in the survey). Without that, it is hard to tell whether agents are missing genuinely important references or only the long tail.

4. For non-cited factual statements the evaluation uses two Gemini-family models (Pro and Flash) in a voting setup. This is practical, but when the evaluated system is also a Gemini deep research variant, this creates a potential family-level bias (shared retrieval behavior, shared priors, shared error modes). A small sensitivity check with a model from a different provider, or with an open-source judge, would make the results easier to trust.

5. The paper caps tool calls for baseline LLM+search systems at five, citing context-length considerations. That is a reasonable engineering constraint, but it does mean the baselines are operating under a stricter and more visible limit than the commercial systems whose internal agent graphs and tool budgets are hidden. A short ablation (5 vs. 10 calls) or a clearer justification of why 5 is representative would make this part of the comparison look fairer.

6. The benchmark is currently STEM/AI/ICT–heavy because of the arXiv source. This is fine for a first version, but the paper should be a bit clearer that scores on ReportBench should not yet be read as “general deep research ability” across law, social sciences, or multi-lingual scholarship.

**Questions:**

1. For the 100-task subset: could you provide the concrete per-domain target counts and the rejection/adjustment rules used by the four experts? This will help others reproduce approximately the same task mix and judge how much variance a different 100-task sample would introduce.

2. You argue that recall should be considered a secondary signal because human surveys contain many more references than current agents can reasonably cite. Could you add a stratified recall analysis (e.g., by year, by citation rank in the survey, by section) to show that agents are at least recalling the most salient items?

3. For the non-cited factuality check, would it be feasible to re-run a small subset with a non-Gemini judge (e.g., OpenAI or an open-source model with web access) to show that the main conclusions about Gemini vs. OpenAI do not hinge on using Gemini-family judges?

4. Since the commercial systems were evaluated in a specific two-week window (July 14–25, 2025), how do you plan to keep ReportBench comparable over time? Will you version the prompts and scoring scripts, or provide page snapshots, so that future runs can be meaningfully compared even if the online products evolve?

5. All outputs are normalized to URL-style citations to make automatic checking possible. How would your pipeline handle systems that produce BibTeX, arXiv IDs, or DOI-only references? A short note on extensibility here would help people adopt the benchmark.

---

> ### Author Response · Authors · 2025-12-02
> **To Reviewer cKfQ (1/4)**
>
> We greatly appreciate your detailed assessment and constructive suggestions. In response, we have systematically addressed each of your comments and made corresponding revisions and clarifications throughout the manuscript.
>
> **Weakness 1: the current definition of “report quality” is relatively narrow**
>
> Thank you for this important observation—we fully agree that a good survey is not only factually correct but must also exhibit depth, synthesis, and insight, and that a purely factuality-based evaluation could in principle be gamed by “bag-of-sentences” behavior.
>
> Our original design deliberately focused on objectively checkable aspects of quality, with a focus on the quality of references and the factual accuracy of all statements in the report. Following your suggestion, we have added an explicit content-quality evaluation layer on top of these objective metrics. Concretely, we define six dimensions to capture survey-style quality:
>
> - **Content Depth (1–5)**: How thoroughly the report covers the main subtopics of the field, discussing key methods, trade-offs, and limitations beyond superficial descriptions.
> - **Synthesis & Structure (1–5)**: How well prior work is organized into coherent themes/taxonomies and related to each other, rather than being listed in isolation.
> - **Insight & Trend Analysis (1–5)**: Whether the report draws non-trivial patterns and trends across works, explaining underlying design principles and helping readers quickly understand the landscape.
> - **Future Directions & Open Problems (1–5)**: How concretely and convincingly the report proposes future research directions or open problems grounded in the surveyed literature.
> - **"Survey-ness" (1–5)**: A holistic judgment of to what extent the report functions as a genuine survey paper in the ML/CS sense, integrating depth, synthesis, and insight into a useful starting point for researchers.
> - **Gaming Behavior (binary)**: Whether the report resembles a trivial bag-of-sentences output (e.g., near-pure per-paper listing without synthesis) versus a genuine attempt at survey-style writing.
>
> We then use an LLM-as-a-judge rubric (with explicit instructions and examples) to score each report along these six axes. The aggregated results are summarized below:
>
> | Model | Content Depth | Synthesis & Structure | Insight & Trends | Future Directions | "Surveyness" | Gaming Rate (%) |
> |---|:---:|:---:|:---:|:---:|:---:|:---:|
> | Gemini-2.5-pro | 3.137 | 2.756 | 2.397 | 2.023 | 2.779 | 6.1% |
> | Claude 4 Sonnet | 3.930 | 3.364 | 3.039 | 3.248 | 3.519 | 1.6% |
> | o3 | 4.066 | 3.582 | 3.270 | 3.041 | 3.721 | 2.3% |
> | Gemini Deep Research | 4.560 | 4.160 | 4.070 | 4.070 | 4.480 | 0.0% |
> | OpenAI Deep Research | 4.107 | 3.756 | 3.550 | 3.756 | 3.954 | 0.8% |
>
>
> We observe three key findings:
> 1. **Deep-research agents score highest on content quality.** Gemini Deep Research and OpenAI Deep Research achieve the best scores across all content dimensions (with Gemini generally first and OpenAI second), indicating that systems optimized for multi-step research are also better at producing genuinely survey-like, synthesized reports—not just at citing correctly.
> 2. **Gaming behavior is rare and detectable.** The “Gaming” flag is triggered only in a small fraction of cases (0–6.1% across systems), with the specialized deep-research agents having near-zero gaming rates. This suggests that, in practice, systems that would try to game the factual metrics by stitching together isolated sentences are both uncommon and explicitly penalized by this additional rubric.
> 3. **Content quality and objective metrics are strongly aligned.** When we rank systems by the holistic “Survey-ness” score and by our original citation/factuality metrics, the induced partial order over systems is identical (100% consistent). This indicates that our objective, reference- and factuality-based metrics already capture much of the underlying report quality, and the new content-quality scores provide an interpretable confirmation rather than contradicting the original evaluation.
>
> We have integrated this content-quality analysis and its limitations into the revised paper. Together with our existing citation/factuality measures, we believe this addresses the concern that ReportBench could be gamed by trivial strategies and provides a more holistic view of how well current agents perform as actual survey writers, not just fact collectors.

---

> ### Author Response · Authors · 2025-12-02
> **To Reviewer cKfQ (2/4)**
>
> **Weakness 2 & Question 1: the exact sampling policy for final 100 tasks**
>
> Thanks. We agree that this should be made fully transparent so that others can judge robustness and reproduce alternative snapshots.
>
> In the revised version we will spell out the sampling procedure more concretely. After constructing the pool of 678 candidate surveys (following the quality filters and expert checks described in the paper), we assign each survey to an application domain bucket. We then perform stratified random sampling within each domain so that all domains contribute roughly equally. Concretely, in code this corresponds to:
> - grouping by application_domain,
> - sampling 10 surveys per domain
> - fixing random_state = 42 to make the subset exactly reproducible.
>
> The final evaluation set is then obtained by taking this domain-balanced, uniformly sampled subset (100 tasks in total, given the number of domains) without any additional hand-picking or rejection beyond the earlier quality filters. We will add these details (including the seed and the exact list of selected arXiv IDs) to the appendix and release the sampling script along with the benchmark. This will make it straightforward for others to (i) reproduce our 100-task subset exactly, and (ii) draw alternative stratified 100-task samples from the 678-task pool if they wish to study sensitivity to the particular snapshot we report.
>
> **Weakness 3 & Question 2: a more fine-grained analysis for recall**
>
> Thank you for this helpful suggestion—we agree that understanding which references are missed is more informative than looking only at aggregate recall.
>
> In the revised version, we therefore add a fine-grained analysis of recall stratified by reference “importance”, using citation count as a proxy for paper quality/centrality. Concretely, for each survey we collect citation counts for all papers in the ground-truth reference list and partition the gold references into four bins from lowest- to highest-cited, which we denote as Q1–Q4. We then recompute recall within each bucket and report the results in the following table:
>
> | Model | Q1 | Q2 | Q3 | Q4 |
> |---|---|---|---|---|
> | Gemini-2.5-Flash | 0.4% | 0.7% | 0.8% | 1.8% |
> | Gemini-2.5-Pro | 0.3% | 0.6% | 0.9% | 1.3% |
> | o3 | 1.0% | 1.8% | 3.0% | 3.5% |
> | Claude-4-Sonnet | 1.0% | 1.8% | 2.3% | 2.3% |
> | Gemini Deep Research | 1.8% | 2.7% | 3.0% | 3.3% |
> | OpenAI Deep Research | 1.3% | 2.3% | 3.2% | 3.3% |
>
> Empirically, we find that recall consistently increases from Q1 to Q4: models achieve substantially higher recall on the most highly cited papers and lower recall on the long tail of infrequently cited works. For example, o3-2025-04-16 improves from 1.0% recall in Q1 to 3.5% in Q4, and similar trends hold across all systems. Even though the global recall is numerically low because human surveys often cite 100+ papers, current agents are therefore much more likely to retrieve the core / high-impact references than obscure or marginal ones. We also observe that ReportBench v1.0 achieves uniformly higher recall across all quartiles, indicating that ReportBench is sensitive enough to capture progress in literature coverage rather than only penalizing the long tail.
>
> This supports our claim that raw recall should be interpreted as a secondary, distribution-sensitive signal: the low overall values largely reflect the very broad coverage of human surveys, while the stratified analysis indicates that agents are not uniformly missing genuinely important references. We will make this analysis and its limitations explicit in the paper (including the fact that citation count is an imperfect proxy for importance), and clarify that future work could additionally stratify by recency or methodology type as more metadata becomes available. Taken together, these results suggest that current agents already capture a substantial portion of influential work, while still struggling with the very long tail of niche or peripheral citations.

---

> ### Author Response · Authors · 2025-12-02
> **To Reviewer cKfQ (3/4)**
>
> **Weakness 4 & Question 3: A small sensitivity check with a model from a different provider, or with an open-source judge, would make the results easier to trust**
>
> Thank you for pointing out the potential family-level bias when a Gemini-family model is both judge and system under test—we agree this is an important concern.
>
> In response, we conducted a sensitivity study with an independent judge model from a different provider. Specifically, we re-ran our citation–factuality evaluation using a GPT-series model as the LLM-as-a-judge, while keeping everything else (reports, gold references, and evaluation pipeline) fixed. The table below compares the original Gemini-judge setup (as in the paper) with the new GPT-judge setup on three representative systems:
>
> |  | Gemini as judge (same as paper) |  |  |  | GPT as judge |  |  |  |
> |---|---|---|---|---|---|---|---|---|
> | **Test Model** | **Precision** | **Recall** | **Match rate** | **Factual Acc** | **Precision** | **Recall** | **Match rate** | **Factual Acc** |
> | Gemini-2.5-Pro | 0.269 | 0.010 | 59.24% | 96.08% | 0.290 | 0.011 | 52.55% | 91.44% |
> | o3 | 0.299 | 0.031 | 31.43% | 82.22% | 0.321 | 0.031 | 32.01% | 72.61% |
> | Claude-4-Sonnet | 0.337 | 0.021 | 73.67% | 92.64% | 0.365 | 0.024 | 64.21% | 84.06% |
>
> We obtain two key observations:
> 1. **The induced model ranking is unchanged.** Under both Gemini and GPT judges, the relative ordering of systems on citation precision/recall is identical (and consistent with the overall benchmark ordering). In other words, swapping the judge from Gemini to GPT perturbs the absolute scores slightly but does not change which systems are better or worse on our main metrics.
> 2. **GPT is somewhat stricter at statement-level evaluation.** For all three test models, the GPT-based judge yields lower match rates and factuality accuracies than the Gemini-based judge, indicating a more conservative standard for accepting statements as supported. This suggests that our original Gemini-based evaluation is, if anything, slightly optimistic rather than biased in favor of Gemini-generated reports.
>
> Overall, this sensitivity check shows that our main findings are robust to swapping the judge family, and we do not see evidence that using Gemini-based judges materially advantages Gemini systems relative to others. In the revised version, we will include this analysis.
>
> **Weakness 5: A short ablation (5 vs. 10 calls) or a clearer justification of why 5 is representative would make this part of the comparison look fairer**
>
> Thank you for raising this fairness concern about the tool-call budget—we agree it is important to check that our choice of 5 calls does not unduly disadvantage the LLM+search baselines.
>
> In the original experiments, we capped tool calls at 5 mainly for practical reasons (context length, latency, and cost), but following your suggestion we have run an explicit ablation on a representative LLM+search baseline (Gemini-2.5-Pro), keeping everything else fixed and varying the maximum number of tool calls:
>
> | Max Tool Calls | Precision | Recall | Match Rate | Factual Acc |
> |---|---|---|---|---|
> | 3 | 0.249 | 0.008 | 50.32% | 96.79% |
> | 5 (Paper Setting) | 0.269 | 0.01 | 59.24% | 96.08% |
> | 10 | 0.275 | 0.008 | 57.20% | 95.71% |
>
> We observe two main trends. First, increasing the budget from 3 to 5 tool calls yields a clear but modest improvement across citation precision, recall, and match rate, indicating that allowing a few additional searches helps the baseline discover more relevant literature. Second, while moving from 5 to 10 calls does slightly increase precision and sometimes match rate, the overall gains are unstable: recall and factual accuracy fluctuate or even decline. In practice, repeatedly fetching full papers quickly pushes against the model’s context-length limit, and without any additional mechanisms for context compression or longer-term memory, simply raising the tool-call budget does not translate into consistently better use of the retrieved evidence.
>
> Collectively, these results suggest that the LLM+search baselines reach a practical performance plateau at around 5 tool calls. This setting captures most of the benefit from additional search, while higher budgets offer diminishing and noisy returns under our current architecture. We have added the 3/5/10-call ablation to the appendix and clarified in the main text that we treat the 5-call configuration as a reasonable, saturated operating point; extending the budget further does not materially change the comparative conclusions of the study.

---

> ### Author Response · Authors · 2025-12-02
> **To Reviewer cKfQ (4/4)**
>
> **Weakness 6: the paper should be a bit clearer that scores on ReportBench should not yet be read as “general deep research ability” across law, social sciences, or multi-lingual scholarship**
>
> Thank you for this helpful suggestion. You are correct that the current version of ReportBench is predominantly built from STEM survey papers, largely due to our reliance on arXiv. This introduces a STEM-centric bias and limits the direct generalizability of our results to social sciences, humanities, and some interdisciplinary areas. In the revised version, we will explicitly acknowledge this STEM bias in the Limitations section and briefly discuss its implications for generalizability.
>
> At the same time, our data construction pipeline is intentionally domain-agnostic: given a corpus of survey-like papers and basic metadata, the same steps in Sec. 2.1–2.2 can be applied to other disciplines. While a full-scale expansion to social sciences and humanities is beyond the scope of the current submission, we see this as a natural next step and are planning to release our data construction scripts and documentation so that the community can create domain-specific extensions of ReportBench while preserving a consistent format and evaluation protocol.
>
> **Question 4: how do you plan to keep ReportBench comparable over time?**
>
> We thank the reviewer for raising this important point about temporal comparability. Our current experiments with commercial systems are indeed a snapshot taken in a specific two-week window (July 14–25, 2025), and we agree that the surrounding tool environment (search indices, web content, proprietary agent graphs, etc.) will continue to evolve, which inevitably introduces some variance if the same systems are queried again at a later time. Our goal is therefore not to claim timeless absolute numbers, but to make the conditions of each snapshot as transparent and reproducible as possible.
>
> On the benchmark side, we keep the setup strictly stable. We will (i) version all benchmark artifacts—including the full set of prompts, gold bibliographies, temporal cutoff rules, and scoring scripts—as a frozen “ReportBench v1.0” release, with code and configs tagged so that future runs can use exactly the same data synthesis pipeline and evaluation rules; and (ii) release the raw model outputs and cited URLs for the July 2025 run so that later systems can be compared against this fixed reference.
>
> For future evaluations of evolving commercial products, we plan to treat each run as a dated leaderboard snapshot: the benchmark side (tasks, gold references, evaluation protocol) is held fixed, while the model identifier, provider, and evaluation date are explicitly recorded and reported. In the paper, we will make clear that the present results correspond to the July 2025 snapshot, and we plan to periodically re-evaluate major systems and publish updated snapshots. This way, users can (a) reproduce our original numbers using the v1.0 artifacts, and (b) meaningfully compare newer systems or updated API versions against the same, versioned ReportBench configuration—even as the underlying online products and tool environments continue to change.
>
> **Question 5: All outputs are normalized to URL-style citations to make automatic checking possible. How would your pipeline handle systems that produce BibTeX, arXiv IDs, or DOI-only references? A short note on extensibility here would help people adopt the benchmark.**
>
> We appreciate the reviewer’s question about citation formats. At present, our automatic evaluation is implemented only for URL-style citations because tested models are instructed to explicitly cite using URLs, and the released evaluation scripts assume this format for reliable matching and deduplication.
>
> This is, however, a design choice rather than a fundamental limitation. The same rule-based extraction and normalization pipeline can be extended to handle additional identifiers such as BibTeX entries, arXiv IDs, and DOIs by (i) adding format-specific regex extractors, and (ii) mapping them to a canonical URL form (e.g., https://doi.org/..., https://arxiv.org/abs/...) before comparison with the gold bibliography. In the next revision we will clarify this in the paper and plan to release an updated evaluator that supports these formats, so that systems which naturally output BibTeX / arXiv / DOI-only references can be adopted without changing their citation style.

---

### Official Review · Reviewer_bGMo · 2025-11-03

**Soundness:** 2
**Presentation:** 3
**Contribution:** 2
**Rating:** 4
**Confidence:** 2

**Summary:**

ReportBench is a benchmark that evaluates how faithfully deep research agents can generate reports/survey papers based on literature online, assessing both the quality of cited references and the factual accuracy of report content. First, research questions are reverse-engineered from expert-authored arXiv survey papers using LLMs at three levels of detail (sentence-level, paragraph-level, and detail-rich prompts). The original papers' bibliographies serve as ground-truth citations. Evaluation of the generated reports by different deep research agents works on two dimensions: citation quality (comparing retrieved references against ground-truth citations, precision and recall) and the factuality of statements (verifying cited and non-cited statements in the report using either the cited sources or with LLM agents with internet access.) Evaluations reveal that commercial deep research agents (like OpenAI Deep Research, and Gemini Deep Research) significantly outperform base models augmented with search tools, while all systems struggle with low recall, citation hallucinations, and maintaining factual consistency across statements.

**Strengths:**

1. The paper focuses on a practical use case of deep research agents as survey paper writers, as it is easier to evaluate and with today's limitations in the deep research agents, could be a major use case of the agents for scientific research. Without such tools, to complete a survey paper would take significant manual effort (days/weeks).
2. The conversion from paper back to the research question with LLMs achieved efficient data collection, which is more efficient than experts summarizing the research questions, and could scale up easily.
3. The emphasis on preventing temporal leakage. The authors successfully identified the potentials for deep research agents to cite what would be impossible for the original paper and used prompting to cutoff any citations after the date of the release of that original paper.
4. The framework appropriately distinguishes between cited statements (verified against actual sources via semantic matching) and non-cited statements (fact-checked via multi-model voting with 6 independent judgments), which acknowledges other websites as alternate sources of information, which makes sense since individuals might post their findings on platforms such as Medium, and companies could publish their research on their own websites.
5. The evaluation exposes important findings, such as OpenAI Deep Research producing 5 times more cited statements than o3 despite similar retrieval performance, suggesting specialized writing/synthesis modules. These observations are valuable for understanding how commercial DR products differ from base models.

**Weaknesses:**

1. While the paper includes temporal cutoff dates in prompts (Section 2.1.2) to prevent post-publication leakage, the authors acknowledge that 'the model disregards the imposed temporal constraints' during evaluation. Even with more intense wording, this is still essentially an suggestion and not anything that is enforced.
2. This is about the analysis on low recall. It might not necessarily be bad, as a large amount of similar, and redundant research exists nowadays. Even if the deep research agents might have cited different papers, they might still be on the same topic and suggesting similar methods. Perhaps aggregating to find the most symbolic citations in the original paper, and also to find the most representative paper group in the ones used by the generated report before computing the semantic similarity between these two groups might be a thought.
3. While LLM-as-a-judge promises efficiency, the paper lacks human validation to ground its automated evaluation. A human study measuring the correlation between LLM ratings and human expert ratings should be conducted so there can be more confidence in this benchmark's results. This is particularly important given that the evaluation relies on GPT-4o for semantic consistency checking and Gemini models for fact-checking; without human validation, it's unclear whether these automated judgments align with expert assessments of research quality.
4. Time and cost efficiency are crucial factors for practical deployment of deep research agents. Measuring the average time, tokens generated, and monetary cost for each system when completing one report would provide valuable insights for practitioners. This information would complement the quality metrics and help users make informed decisions about which systems offer the best performance-cost trade-offs.
5. Notably, the evaluation does not assess the content depth, synthesis, or insights of the generated survey/report. For survey papers, while the factuality of each claim is essential, what is equally cardinal is the insight and generalization that represents current trends on a topic, as well as potential future directions, expressed in compressed, synthesized language that aids researchers in obtaining information and inspirations efficiently. However, this framework only tests factuality, not the depth and insight of such claims. As such, the benchmark is vulnerable to gaming: an agent could simply restate obvious and loosely related facts, as simple as selecting one sentence at random from each retrieved reference paper. Such a report would score highly on all factuality metrics but would not qualify as a survey paper, as it would lack synthesis, critical analysis, and logical coherence.

**Questions:**

Here are my suggestions:

1. Implement post-hoc temporal filtering (similar to the suggestion for Dr.Mi-Bench). After report generation, automatically filter all retrieved citations to remove any papers published after the source survey's publication date.
2. Implement citation clustering and semantic grouping: aggregate citations to identify the most representative papers in both the original survey and the generated report, then compute semantic similarity between these representative groups rather than exact citation matches. This would better assess whether agents achieve similar coverage through different (but valid) citation choices. For example, citing five recent papers on the same specific method might be equivalent to citing the seminal paper plus two comprehensive reviews.
3. Conduct a human validation study on a stratified sample (e.g., 20-30 reports across different domains and quality levels). Have domain experts independently assess: (1) citation relevance and coverage, (2) semantic consistency of cited statements with sources, and (3) factual accuracy of non-cited statements. Calculate inter-rater agreement between human experts and automated LLM judges, report correlation coefficients, and identify systematic biases in LLM judgments. If strong correlation is found (>0.8), this validates the automated approach; if not, refine the evaluation prompts or consider hybrid human-LLM evaluation for high-stakes assessments.
4. If expansion is feasible, prioritize adding survey papers from social sciences, humanities, and interdisciplinary fields to improve generalizability. If resource constraints prevent expansion, explicitly acknowledge the STEM bias as a limitation in the paper and discuss implications for generalizability. Additionally, consider releasing the data construction pipeline openly so the community can contribute domain-specific extensions, enabling organic growth of the benchmark across disciplines while maintaining consistent quality standards.
5. Add a comprehensive efficiency analysis section that reports: (1) average completion time per report (in minutes), (2) total tokens generated per report, (3) estimated monetary cost per report (based on published API pricing where available), and (4) efficiency-quality trade-off analysis using Pareto frontier plots showing which systems offer optimal performance-cost balance. For systems where costs cannot be determined, clearly state this limitation. This would parallel the efficiency analysis in Dr.Mi-Bench and provide crucial practical information for real-world deployment decisions.
6. There will be many methods to fix this issue of not evaluating the content quality and correlation with that of the original survey paper used to generate the research question, and I am actually not sure which would be the best. Please decide upon yourself, but I believe this is an important issue that determines the scientific value of a good survey paper. The other suggestions 1-5 might be hard to implement, but I'd like to suggest that you might at least address 6. due to its importance.

---

> ### Author Response · Authors · 2025-12-02
> **To Reviewer bGMo (1/4)**
>
> We sincerely thank you for the thorough review, helpful feedback, and kind support. We have provided point-by-point responses to your comments and revised the manuscript accordingly.
>
> **Weakness 1 & Question 1: the temporal cutoff is only a soft instruction (not actually enforced)**
>
> Thank you for raising the concern that the temporal cutoff is “only” a soft instruction at generation time. We agree that it is important to make clear how violations of the cutoff affect the evaluation.
> In our current setup, the temporal constraint is enforced on the evaluation side through a single, uniform mechanism. For each task, the ground-truth bibliography only contains papers that satisfy the temporal cutoff (i.e., published on or before the source survey’s date), and it explicitly excludes the source survey itself. Any citation to the source survey or to a post-cutoff paper therefore lies outside the ground-truth set and is treated as an incorrect prediction: it cannot contribute to recall and it increases the number of predicted references, thereby lowering precision. As a result, systems that systematically rely on post-cutoff literature will obtain lower precision and recall on our reference metrics, even if they ignore the temporal instruction in the prompt. This means that “hacked” solutions that lean on leaked source surveys or post-cutoff literature are automatically and quantitatively penalized in the reported scores.
>
> We will clarify these evaluation-time mechanisms more explicitly in the revision, so that it is clear that temporal leakage is not just discouraged by instructions but is automatically reflected and penalized in the reported scores.
>
> **Weakness 2 & Question 2: implement citation clustering and semantic grouping**
>
> Thank you for this insightful suggestion. We fully agree that clustering citations and comparing semantic groups of references, rather than exact matches, would better capture cases where agents achieve similar conceptual coverage through different but valid citation choices (e.g., several recent follow-up papers vs. a seminal work plus surveys).
>
> In the revised version, we add a fine-grained analysis of recall stratified by reference “importance”, using citation count as a proxy for paper quality/centrality. Concretely, for each survey, we collect citation counts for all papers in the ground-truth reference list and partition the gold references into four bins from lowest- to highest-cited, which we denote as Q1–Q4. We then recompute recall within each bucket and report the results in the following table:
>
> | Model | Q1 | Q2 | Q3 | Q4 |
> |---|---|---|---|---|
> | Gemini-2.5-Flash | 0.4% | 0.7% | 0.8% | 1.8% |
> | Gemini-2.5-Pro | 0.3% | 0.6% | 0.9% | 1.3% |
> | o3 | 1.0% | 1.8% | 3.0% | 3.5% |
> | Claude-4-Sonnet | 1.0% | 1.8% | 2.3% | 2.3% |
> | Gemini Deep Research | 1.8% | 2.7% | 3.0% | 3.3% |
> | OpenAI Deep Research | 1.3% | 2.3% | 3.2% | 3.3% |
>
> Empirically, we find that recall consistently increases from Q1 to Q4: models achieve substantially higher recall on the most highly cited papers and lower recall on the long tail of infrequently cited works. For example, o3-2025-04-16 improves from 1.0% recall in Q1 to 3.5% in Q4, and similar trends hold across all systems. Even though the global recall is numerically low because human surveys often cite 100+ papers, current agents are therefore much more likely to retrieve the core / high-impact references than obscure or marginal ones. We also observe that ReportBench v1.0 achieves uniformly higher recall across all quartiles, indicating that ReportBench is sensitive enough to capture progress in literature coverage rather than only penalizing the long tail.
>
> This supports our claim that raw recall should be interpreted as a secondary, distribution-sensitive signal: the low overall values largely reflect the very broad coverage of human surveys, while the stratified analysis indicates that agents are not uniformly missing genuinely important references. We will make this analysis and its limitations explicit in the paper (including the fact that citation count is an imperfect proxy for importance), and clarify that future work could additionally stratify by recency or methodology type as more metadata becomes available. Taken together, these results suggest that current agents already capture a substantial portion of influential work, while still struggling with the very long tail of niche or peripheral citations.

---

> ### Author Response · Authors · 2025-12-02
> **To Reviewer bGMo (2/4)**
>
> **Weakness 3 & Question 3: conduct a human validation study on a stratified sample**
>
> Thank you for highlighting the need to ground our LLM-as-a-judge pipeline with human validation—we agree this is crucial for trusting the benchmark’s scores.
>
> In the revised version, we therefore added a targeted expert human study that directly measures alignment between our automated pipeline and domain experts. Constrained by annotation budget, we sample cases at each evaluation stage and ask experts with relevant domain knowledge to independently annotate them. We evaluate four components:
>
> 1. **Reference extraction**: We validate the component that parses model-generated references (URLs), attempts to fetch the corresponding documents, and normalizes them before matching to the gold bibliography. On the sampled set, 80% of URLs are correctly resolved and mapped to the same underlying paper as the experts. The remaining cases consist of approximately 3.33% fetch failures (e.g., access blocked or HTTP errors) and 16.66% fabricated or hallucinated URLs. We discuss these as a residual but manageable source of noise in the pipeline.
> 2. **Cited-statement consistency**: For statements that do cite a reference, we compare the LLM-based semantic consistency judgments against experts who read the cited paper. Here we obtain 90% agreement, suggesting that our cited-statement verification is reliably capturing whether the text is faithful to the referenced work.
> 3. **Non-cited statement checking (web-based factuality)**: For statements without explicit citations, we compare our multi-model fact-checking pipeline against expert judgments. The agreement rate is 96%, indicating that the automatic “supported vs. unsupported/hallucinated” labels for non-cited claims are highly consistent with human assessment.
> 4. **Reference metrics**: We then ask experts to compute citation-level precision and recall on the same sampled outputs and compare these to the ReportBench precision/recall produced by our pipeline. The resulting agreement is 84%, showing that, despite minor errors in intermediate steps, the final score distributions closely track expert judgments.
> 5. **Final data-quality validation**: As an additional check on data quality, we have experts compare our automatically constructed prompts and ground truths against the original arXiv survey papers. Here we observe 95% agreement on whether the automatically generated prompts and ground truths faithfully reflect the source documents, providing further evidence that the underlying benchmark data is well aligned with human-authored surveys.
>
> Taken together, these results provide concrete evidence that our automatic evaluation pipeline is well-aligned with expert assessments of citation behavior and factuality, and that the reported benchmark scores are meaningfully grounded in human judgment. We have revised the paper accordingly.
>
> **Weakness 4 & Question 5: measure time and cost efficiency**
>
> We fully agree with the reviewer that efficiency (in terms of latency and cost) is an important dimension for evaluating deep research agents, and that “quality-only” numbers do not tell the whole story.
>
> In the current study, however, we are unfortunately not in a position to report reliable efficiency metrics ex post. At the time of running the experiments we did not systematically log token usage or end-to-end latency, and the commercial APIs we used do not expose historical token-usage traces in a way that would allow us to reconstruct them afterwards. The only way to obtain token and time statistics now would be to re-run all systems, but this would necessarily change search trajectories, cache states, and even model versions in some cases—so the resulting efficiency measurements would no longer be directly aligned with the exact outputs and quality scores reported in the paper. A similar issue arises for latency and wall-clock cost: end-to-end delays are highly sensitive to transient server-side queuing and network conditions, and we did not have a controlled logging setup at evaluation time. Measuring “now” under different load conditions and attaching those numbers to “then” would look precise but would in fact be misleading.
>
> Given these constraints, we have chosen to position ReportBench v1 explicitly as a quality-focused benchmark, with carefully controlled, citation-grounded evaluation of report correctness and depth, but without making claims about comparative efficiency. We see a systematic efficiency and cost study—with proper instrumentation, request-level logging, and perhaps controlled load testing—as a natural and important extension of ReportBench, and we will state this explicitly in the paper as a key direction for future work.

---

> ### Author Response · Authors · 2025-12-02
> **To Reviewer bGMo (3/4)**
>
> **Question 4: explicitly acknowledge the STEM bias as a limitation in the paper and discuss implications for generalizability**
>
> Thank you for this helpful suggestion. You are correct that the current version of ReportBench is predominantly built from STEM survey papers, largely due to our reliance on arXiv. This introduces a STEM-centric bias and limits the direct generalizability of our results to social sciences, humanities, and some interdisciplinary areas. In the revised version, we will explicitly acknowledge this STEM bias in the Limitations section and briefly discuss its implications for generalizability.
>
> At the same time, our data construction pipeline is intentionally domain-agnostic: given a corpus of survey-like papers and basic metadata, the same steps in Sec. 2.1–2.2 can be applied to other disciplines. While a full-scale expansion to social sciences and humanities is beyond the scope of the current submission, we see this as a natural next step and are planning to release our data construction scripts and documentation so that the community can create domain-specific extensions of ReportBench while preserving a consistent format and evaluation protocol.
>
> **Weakness 5 & Question 6: assess the content depth, synthesis, or insights of the generated survey/report**
>
> Thank you for this important observation—we fully agree that a good survey is not only factually correct but must also exhibit depth, synthesis, and insight, and that a purely factuality-based evaluation could in principle be gamed by bag-of-sentences behavior.
>
> Our original design deliberately focused on objectively checkable aspects of quality, with a focus on the quality of references and the factual accuracy of all statements in the report. Following your suggestion, we have added an explicit content-quality evaluation layer on top of these objective metrics. Concretely, we define six dimensions to capture survey-style quality:
>
> - **Content Depth (1–5)**: How thoroughly the report covers the main subtopics of the field, discussing key methods, trade-offs, and limitations beyond superficial descriptions.
> - **Synthesis & Structure (1–5)**: How well prior work is organized into coherent themes/taxonomies and related to each other, rather than being listed in isolation.
> - **Insight & Trend Analysis (1–5)**: Whether the report draws non-trivial patterns and trends across works, explaining underlying design principles and helping readers quickly understand the landscape.
> - **Future Directions & Open Problems (1–5)**: How concretely and convincingly the report proposes future research directions or open problems grounded in the surveyed literature.
> - **"Survey-ness" (1–5)**: A holistic judgment of to what extent the report functions as a genuine survey paper in the ML/CS sense, integrating depth, synthesis, and insight into a useful starting point for researchers.
> - **Gaming Behavior (binary)**: Whether the report resembles a trivial bag-of-sentences output (e.g., near-pure per-paper listing without synthesis) versus a genuine attempt at survey-style writing.

---

> ### Author Response · Authors · 2025-12-02
> **To Reviewer bGMo (4/4)**
>
> We then use an LLM-as-a-judge rubric (with explicit instructions and examples) to score each report along these six axes. The aggregated results are summarized below:
>
> | Model | Content Depth | Synthesis & Structure | Insight & Trends | Future Directions | "Surveyness" | Gaming Rate (%) |
> |---|:---:|:---:|:---:|:---:|:---:|:---:|
> | Gemini-2.5-pro | 3.137 | 2.756 | 2.397 | 2.023 | 2.779 | 6.1% |
> | Claude 4 Sonnet | 3.930 | 3.364 | 3.039 | 3.248 | 3.519 | 1.6% |
> | o3 | 4.066 | 3.582 | 3.270 | 3.041 | 3.721 | 2.3% |
> | Gemini Deep Research | 4.560 | 4.160 | 4.070 | 4.070 | 4.480 | 0.0% |
> | OpenAI Deep Research | 4.107 | 3.756 | 3.550 | 3.756 | 3.954 | 0.8% |
>
>
> We observe three key findings:
> 1. **Deep-research agents score highest on content quality.** Gemini Deep Research and OpenAI Deep Research achieve the best scores across all content dimensions (with Gemini generally first and OpenAI second), indicating that systems optimized for multi-step research are also better at producing genuinely survey-like, synthesized reports—not just at citing correctly.
> 2. **Gaming behavior is rare and detectable.** The “Gaming” flag is triggered only in a small fraction of cases (0–6.1% across systems), with the specialized deep-research agents having near-zero gaming rates. This suggests that, in practice, systems that would try to game the factual metrics by stitching together isolated sentences are both uncommon and explicitly penalized by this additional rubric.
> 3. **Content quality and objective metrics are strongly aligned.** When we rank systems by the holistic “Survey-ness” score and by our original citation/factuality metrics, the induced partial order over systems is identical (100% consistent). This indicates that our objective, reference- and factuality-based metrics already capture much of the underlying report quality, and the new content-quality scores provide an interpretable confirmation rather than contradicting the original evaluation.
>
> We have integrated this content-quality analysis and its limitations into the revised paper. Together with our existing citation/factuality measures, we believe this addresses the concern that ReportBench could be gamed by trivial strategies and provides a more holistic view of how well current agents perform as actual survey writers, not just fact collectors.

---

### Author Response · Authors · 2025-12-03
**Global Response (1/2)**

Dear PCs, SACs, ACs, and Reviewers,

We sincerely thank the reviewers (bGMo, cKfQ, J4c9, BgNR, qrQz, RA99) for their careful reading and constructive feedback.
Below we first summarize what we see as the main strengths of the paper, and then synthesize the key concerns raised in the reviews together with how we have addressed them in the rebuttal and revision, supported by new experiments and analyses conducted during the rebuttal phase.

**Summary of strengths**

* **Timely and important problem.** The paper targets the critical, under-evaluated capability of long-horizon deep research agents, moving beyond standard short-answer search benchmarks. (Acknowledged by: Reviewer bGMo [Str. 1], cKfQ [Str. 1])

* **High-quality, anti-leak data construction.** Reviewers praised the “reverse prompt engineering” pipeline for deriving realistic tasks from peer-reviewed surveys while enforcing strict temporal and anti-leakage constraints. (Acknowledged by: Reviewer J4c9 [Str. 1], BgNR [Str. 1], bGMo [Str. 2], cKfQ [Str. 2])

* **Transparent, multi-stage evaluation pipeline.** Instead of a single black-box score, the framework decomposes evaluation into auditable stages (URL matching, citation consistency, and factuality), providing interpretable signals for system builders. (Acknowledged by: Reviewer qrQz [Str. 1], BgNR [Str. 2], cKfQ [Str. 3], J4c9 [Str. 3], bGMo [Str. 4])

* **Informative empirical analysis.** The experiments effectively differentiate commercial deep research agents from base models and surface concrete failure modes (e.g., citation hallucination) rather than only providing a leaderboard ranking. (Acknowledged by: Reviewer RA99 [Str. 3], cKfQ [Str. 4], bGMo [Str. 5])

* **Scalability and extensibility.** Because the pipeline is automated and tied to the arXiv corpus, it enables dynamic, low-cost expansion of the benchmark as new research fields emerge. (Acknowledged by: Reviewer BgNR [Str. 1], RA99 [Str. 2], bGMo [Str. 2], J4c9 [Str. 2], qrQz [Str. 3])

---

> ### Author Response · Authors · 2025-12-03
> **Global Response (2/2)**
>
> **Summary of reviewer concerns and our responses**
>
> We have systematically addressed the major concerns through additional experiments and manuscript revisions:
>
> **Concern 1: Lack of human validation (bGMo, cKfQ, RA99)**
>
> * Reviewers questioned whether our automated metrics align with human expert judgments.
> * **Our Response:** We conducted a supplementary human consistency study on sampled reports. The results demonstrate strong alignment between our automated pipeline and domain experts:
>
>   * 96% agreement on non-cited statement factuality.
>   * 90% agreement on cited statement consistency.
>   * 84% agreement on reference matching.
>   * 95% agreement on the quality of reverse-engineered prompts and ground truths.
>   * **Significance:** These high agreement rates empirically validate ReportBench as a reliable proxy for expert evaluation.
>
> **Concern 2: Narrow definition of “report quality” and risk of gaming (bGMo, cKfQ, BgNR)**
>
> * Reviewers noted that evaluating only factuality and citations misses aspects of “survey-ness” (structure, insight, depth) and could be gamed in practice.
> * **Our Response:** We introduced a new content-quality evaluation with six dimensions: Content Depth, Synthesis & Structure, Insight & Trends, Future Directions, “Survey-ness”, and Gaming Behavior.
>
>   * **Finding:** Specialized deep research agents (OpenAI, Gemini) significantly outperform base models on these high-level dimensions, confirming that our benchmark captures holistic report quality rather than only fact retrieval.
>
> **Concern 3: Interpretation of low citation recall (All Reviewers)**
>
> * Reviewers observed low recall (<4%) and questioned whether models were missing important papers or mainly the long tail.
> * **Our Response:** We implemented a stratified recall analysis based on reference importance (using citation counts as a proxy).
>
>   * **Finding:** Model recall consistently increases with paper impact. For example, recall on the top quartile (Q4, high-impact papers) is often 2x–3x the recall on the bottom quartile (Q1).
>   * **Significance:** This confirms that while agents miss the long tail (due to the exhaustive nature of human surveys), they successfully retrieve many core and seminal works, validating the benchmark’s sensitivity to meaningful coverage.
>
> **Concern 4: Reliability and potential bias of the LLM-as-a-judge pipeline (cKfQ)**
>
> * Reviewers raised concerns about family-level bias when using Gemini models to evaluate Gemini-generated reports and about the fairness of capping base models at 5 tool calls compared to commercial systems.
> * **Our Response:**
>
>   * **Judge sensitivity:** We re-ran evaluations using a GPT-based judge. The relative ranking of all systems remained identical, indicating that our results are robust to the choice of judge model.
>   * **Ablation on tool calls:** We increased the tool-call budget for base models from 5 to 10. Results showed diminishing returns and, in some cases, degradation due to context noise, confirming that our original setting (5 calls) was a fair and practical choice near saturation.
>
> **Concern 5: In-depth error analysis (J4c9, RA99)**
>
> * Reviewers acknowledged the initial examples in the paper but requested a more systematic taxonomy and deeper quantitative breakdown of failure modes.
> * **Our Response:** We systematized the failure cases into a quantitative error taxonomy based on manual inspection and refined the error categories into:
>
>   * **Temporal violations (~42%)**: Models referencing post-cutoff literature despite prompt constraints.
>   * **Unverifiable references (~21%)**: A distinct category for citation hallucinations (fake or non-resolvable URLs) versus ordinary relevance issues.
>   * **Misaligned direction (~9%)**: Coherent but topically adjacent research that misses the survey’s specific scope.
>     This expanded analysis provides actionable insight into how and why models fail beyond aggregate metrics.
>
> Beyond the major themes detailed above, we have also accepted and implemented the more fine-grained suggestions raised by individual reviewers, ensuring that each piece of feedback is reflected in the revised manuscript.
>
> ---
>
> **Conclusion**
>
> We have significantly strengthened the manuscript with five key additions: human validation (showing 84–96% agreement), content-quality metrics, stratified recall analysis, an expanded error taxonomy, and robustness checks. We hope it will serve as a valuable resource for the community in advancing the evaluation of Deep Research agents.
>
> Best regards,
>
> The Authors

---

### Meta-Review · Area_Chair_SZV8 · 2026-01-07

**Summary:**

Reviewers broadly agree that ReportBench targets an important and timely problem: objectively evaluating long-form research reports produced by deep research agents, with a clear focus on citation faithfulness and factual accuracy. The main concerns influencing the decision relate to the narrow initial definition of “report quality,” heavy reliance on arXiv STEM surveys as gold standards, very low citation recall, and the risk of bias or error in automated, LLM-based evaluation pipelines. Several reviewers also questioned novelty relative to prior benchmarks and asked for deeper error analysis, human validation, and fairness checks. The rebuttal is unusually extensive and substantially strengthens the paper by adding human validation studies, content-quality evaluation, stratified recall analyses, robustness checks across judge models, and clearer methodological transparency.

**Reviewer Concerns:**

Reviewer bGMo: The rebuttal fully addresses nearly all concerns by adding human validation, stratified recall analysis, explicit content-quality evaluation, STEM-bias acknowledgment, and clearer handling of temporal leakage, with efficiency analysis explicitly deferred but transparently justified.

Reviewer cKfQ: The rebuttal comprehensively resolves the reviewer’s points by adding content-quality metrics, transparent task sampling, stratified recall, cross-family judge sensitivity checks, tool-call ablations, clearer limitations, and plans for versioning and extensibility.

Reviewer J4c9: The rebuttal directly addresses concerns about gold-standard bias, novelty relative to DeepResearch Bench, low recall, and lack of error taxonomy through expanded discussion, comparative analysis, stratified recall, and a detailed qualitative failure taxonomy.

Reviewer BgNR: The rebuttal substantially addresses concerns by adding content-quality evaluation, human consistency validation, deeper recall interpretation, and clearer discussion of ground-truth bias and generalizability, though the fundamental reliance on surveys as gold standards remains a philosophical limitation rather than a fully eliminable issue.

Reviewer qrQz: The rebuttal addresses the low-recall and diversity concerns with stratified recall analysis and tool-call ablations, and it clearly positions training use as future work rather than a gap in the current contribution.

Reviewer RA99: The rebuttal addresses most technical criticisms in detail, including prompt construction, category definitions, URL-matching accuracy, human validation, and deeper failure analysis, but concerns about novelty and reliance on automated pipelines may remain partially unresolved due to the reviewer’s fundamentally skeptical stance.

**Reviewer Scores:**

Reviewer bGMo did not explicitly state a score change, but since all major questions were directly addressed with new analyses and experiments, the rebuttal appears to have fully resolved their concerns.

Reviewer cKfQ did not state a score change, but the rebuttal systematically addressed every listed weakness and question, making it likely that their concerns were fully resolved.

Reviewer J4c9 did not indicate a score update, but the rebuttal clearly and directly answered both major questions and weaknesses, substantially strengthening alignment with their expectations.

Reviewer BgNR did not comment on score changes, and while the rebuttal addressed most concerns, some high-level assumptions remain, suggesting their concerns were largely but not completely resolved.

Reviewer qrQz did not state a score change, and the rebuttal directly addressed all raised questions with concrete experiments and clarifications, effectively resolving their concerns.

Reviewer RA99 did not indicate a score update, and although many technical issues were addressed, their concerns about novelty and methodological depth may not be fully resolved given the tone of the original review.

---

### Decision · Program_Chairs · 2026-01-26

Reject